# Relating gut microbiome composition and life history metrics for pronghorn (*Antilocapra americana*) in the Red Desert, Wyoming

Courtney E. Buchanan[1]*, Stephanie J. Galla[2], Mario E. Muscarella[3], Jennifer S. Forbey[2], Adele K. Reinking[1,4,5,6], Jeffrey L. Beck[1]

1 Department of Ecosystem Science and Management, University of Wyoming, Laramie, Wyoming, United States of America, 2 Department of Biological Sciences, Boise State University, Boise, Idaho, United States of America, 3 Institute of Arctic Biology and Department of Biology & Wildlife, University of Alaska Fairbanks, Fairbanks, Alaska, United States of America, 4 Cooperative Institute for Research in the Atmosphere, Colorado State University, Fort Collins, Colorado, United States of America, 5 Graduate Degree Program in Ecology, Colorado State University, Fort Collins, Colorado, United States of America, 6 Department of Fish, Wildlife, and Conservation Biology, Colorado State University, Fort Collins, Colorado, United States of America

* cbuchan3@uwyo.edu

**Data Availability Statement:** Sequence data are stored on the Qiita platform under study ID 12842 which has been made publicly available to anyone

## Abstract

Host microbial communities (hereafter, the 'microbiome') are recognized as an important aspect of host health and are gaining attention as a useful biomarker to understand the ecology and demographics of wildlife populations. Several studies indicate that the microbiome may contribute to the adaptive capacity of animals to changing environments associated with increasing habitat fragmentation and rapid climate change. To this end, we investigated the gut microbiome of pronghorn (*Antilocapra americana*), an iconic species in an environment that is undergoing both climatic and anthropogenic change. The bacterial composition of the pronghorn gut microbiome has yet to be described in the literature, and thus our study provides important baseline information about this species. We used 16S rRNA amplicon sequencing of fecal samples to characterize the gut microbiome of pronghorn—a facultative sagebrush (*Artemisia spp.*) specialist in many regions where they occur in western North America. We collected fecal pellets from 159 captured female pronghorn from four herds in the Red Desert of Wyoming during winters of 2013 and 2014. We found small, but significant differences in diversity of the gut microbiome relative to study area, capture period, and body fat measurements. In addition, we found a difference in gut microbiome composition in pronghorn across two regions separated by Interstate 80. Results indicated that the fecal microbiome may be a potential biomarker for the spatial ecology of free-ranging ungulates. The core gut microbiome of these animals—including bacteria in the phyla Firmicutes (now Bacillota) and Bacteroidota—remained relatively stable across populations and biological metrics. These findings provide a baseline for the gut microbiome of pronghorn that could potentially be used as a target in monitoring health and population structure of pronghorn relative to habitat fragmentation, climate change, and management practices.

who creates a Qiita account. https://qiita.ucsd.edu/study/description/12842. This data has also been placed into the EBI-ENA repository and is publicly available under accession number PRJEB68147. Code for the QIIME2 pipeline we used along with metadata and R code for downstream data analysis are available at https://github.com/courtney-buchanan/Pronghorn_microbiome. The metadata file used for R analysis is also included in the Supporting information.

**Funding:** "The sequencing data collection and analysis and preparation of the manuscript was supported by the U.S. National Science Foundation (https://www.nsf.gov/): Grants OIA-1826801 to J. Forbey and OIA-1738865 to E. Hayden. Funding for fecal sample and pronghorn field data collection was funded with a combination of grants and financial support from agency, university, and industry partners listed below. Funds were awarded to J. Beck between 2009 and 2016 for previous pronghorn research. Our research used legacy samples that were collected during this previous research project. Anadarko Petroleum Corporation (acquired by Occidental Petroleum in 2019 https://www.oxy.com/). Institution award number: 1002418 Black Diamond Minerals LLC (Merged into MRD Operating, LLC in 2014). Institution award number: 1001671 British Petroleum North America (https://www.bp.com/). No institution award number or grant information available. Bureau of Land Management-Rawlins Field Office (https://www.blm.gov/office/rawlins-field-office). Sponsor ID: L16AC00156 and L09AC15996 Devon Energy (https://www.devonenergy.com/). Institution award number: 1002087 Linn Energy (https://linnenergy.com/). Institution award number: 1002724 Memorial Resource Development (acquired by Range Resources Corporation in 2016 https://www.rangeresources.com/). Institution award number: 1002818 Samson Resources (https://www.samsonco.com/index.aspx). Institution award number: 1001972 Warren Resources, Incorporated (https://warrenresources.com). No institution award number or grant information available. Wyoming Game and Fish Department (https://wgfd.wyo.gov/). No institution award number or grant information available. Wyoming Governor's Big Game License Coalition (https://wgfd.wyo.gov/Apply-or-Buy/Commissioner-and-Governor-Licenses/Governors-Big-Game-License). No institution award number or grant information available. University of Wyoming: Department of Ecosystem Science and Management (https://www.uwyo.edu/esm/index.html), Office of Academic Affairs (https://www.uwyo.edu/acadaffairs/index.html), and Wyoming Reclamation

## Introduction

Species do not exist in isolation, but rather experience interactions with a myriad of other species, including microorganisms. In a growing number of publications, individuals are viewed as a holobiont—a combination of the host and associated microbes, rather than a standalone organism (reviewed in [1]). Holobionts possess a hologenome, which is the sum of the host and microbial genomes [1]. Understanding how organisms function as a holobiont underpins symbiotic relationships that contribute to host physiology and demographics. The composition of the microbiome can influence many bodily processes, including immunity (reviewed in [2, 3]) and reproduction (reviewed in [4]). Specifically, the gut microbiome- or the microorganisms living in the digestive tract of an animal host—can have a large influence on processes related to digestion and nutrient absorption. To illustrate, the importance of symbiotic microbial genomes for cellulose digestion in mammalian herbivores has long been known [5, 6]. Growing research has recently revealed other important health effects of microbial symbionts. For example, the gut microbiome has been linked to feed efficiency in livestock [7–9], and studies in both house mice (*Mus musculus*) and humans (*Homo sapiens*) have established links between gut microbiome composition and fat deposition, body condition, and metabolism [10–15]. Microbes also may grant host species the ability to degrade secondary metabolites in plant foods, as seen in greater sage-grouse (*Centrocercus urophasianus* [16]), several insect species (see examples in [17]), and woodrats (*Neotoma lepida* [18]).

While host-microbiome relationships are known to influence host health in humans, livestock, and model organisms and can potentially influence the management of wildlife species worldwide [19], fewer studies have investigated the influence of the microbiome on the health of wild animals [20]. Wild animals are more difficult to study because they often inhabit remote locations, and study conditions are difficult to control relative to laboratory environments. However, wild animals have been shown to possess distinct gut microbiomes from their captive or domestic counterparts [21–25]. As native landscapes undergo increasing conversion, fragmentation, and rapid climate change, microbial plasticity may confer greater adaptive capacity among host animals to mitigate deleterious effects [19, 26, 27]. Although common in humans and agricultural applications, probiotic treatments may also represent a management tool for wildlife species [19, 28]. Additionally, recent research has explored topical applications of microbes to fight infectious diseases in bats [29] and amphibians [30] and evaluated the potential to use other strains of bacteria as wildlife gut probiotics [31]. Studies of host-microbiome interactions in wildlife could prove informative, particularly in habitats undergoing a change in land use.

Rangelands in the Intermountain West—which are dominated by sagebrush (*Artemisia spp.*)—serve as an ideal location to study wild microbiomes and their effects on host health in a rapidly changing landscape. The sagebrush steppe ecosystem covers a large portion of terrestrial North America. However, it has been reduced to 56% of its historic extent due to anthropogenic development, conversion to cropland, invasion of non-native plants, and conifer encroachment [32, 33]. Sagebrush species along with other woody plants in this ecosystem (e.g., bitterbrush [*Purshia tridentata*], rabbitbrush [*Chrysothamnus spp.*], greasewood [*Sarcobatus vermiculatus*], saltbushes [*Atriplex spp.*], and junipers [*Juniperus spp.*]) are defended with potentially toxic chemicals [34–42]. Greater and Gunnison sage-grouse (*Centrocercus spp.*), pygmy rabbits (*Brachylagus idahoensis*), and pronghorn (*Antilocapra americana*) represent the relatively few vertebrate herbivores able to consume large amounts of sagebrush. Sagebrush can comprise nearly 100% of the diet of pygmy rabbits and sage-grouse in winter [43, 44]. These species have known physiological adaptations that explain their tolerance to sagebrush toxins [44–47] including the gut microbiota in sage-grouse that may degrade toxins

and Restoration Center (https://www.uwyo.edu/wrrc/index.html). No institution award number or grant information available. Funders did not play a role in study design, data collection and analysis, decision to publish, or preparation of the manuscript with the exception of logistical and field support provided by BLM and WGFD staff mentioned in acknowledgements during the sample collection phase of the project."

**Competing interests:** The authors have declared that no competing interests exist.

[16]. While sagebrush may also dominate the diets of pronghorn [48–51], they can also subsist on grasses and forbs [51, 52]. Pronghorn can also shift to entirely different chemically-defended shrubs (e.g., rabbitbrush) when habitat fragmentation and degradation alters shrub communities [53]. Unlike other sagebrush specialists, the ruminant digestion [54], migratory behavior [55–57], and observed dietary plasticity [53] of pronghorn may result in unique microbial communities and adaptations.

As landscapes continue to change, there is a need to understand the relationships between hosts, their microbial symbionts, and the environment. Here, we studied the gut microbiome of pronghorn, an endemic and iconic big game species of the American West with considerable ecological and economic value that reside largely in the sagebrush steppe. While the species' population size is lower than historical levels (i.e., before westward expansion), there are around 900,000 pronghorn as of 2017, and recent population trends are stable or increasing [58, 59]. However, some areas have seen local declines in pronghorn populations [58, 60–62]. Several anthropogenic factors create disturbance that pronghorn avoid and sometimes increase mortality for pronghorn, including fencing [63–68], livestock agriculture [57], human development [69], roads [67–72], and energy development [64, 67, 72–74]. Environmental factors such as harsh winters or climatic changes [61, 62, 75, 76], disease [77], and coyote (*Canis latrans*) predation of fawns [78] can also negatively affect pronghorn populations. Epizootic Hemorrhagic Disease (EHD) and Bluetongue Virus (BTV) are two hemorrhagic diseases of pronghorn and other ungulates that can cause large die off events of animals [77, 79, 80], with one outbreak of BTV killing an estimated 3,200 pronghorn [77]. Pronghorn with higher body condition scores have been shown to be more resilient to harsh winters [75], and populations in better overall health and body condition likely will be more resilient to multiple stressors. Body condition has been positively related to population growth in bighorn sheep (*Ovis canadensis* [81]) and mule deer (*Odocoileus hemionus* [82]). Because gut microbiome composition has been related to feed efficiency and various measures of production in livestock [7–9], it is possible that gut microbiome composition could serve as a potential bio-indicator in wildlife populations as well. Indeed, work in mule deer has shown relationships between specific bacteria taxa and health metrics relating to protein and fat storage [83].

With the exception of studies evaluating protozoa in the rumen [84] and gut anaerobic fungi [85], the pronghorn gut microbiome has not been studied. To our knowledge, our study is the first to characterize bacterial gut microbiome communities in pronghorn. Here, we use fecal samples to describe the gut microbiome community in pronghorn and provide novel, baseline information on the core bacterial gut microbiome that can be used in future studies to compare with pronghorn residing in different environments, during different seasons, or after experiencing predicted future climatic or anthropogenic changes. In addition to this objective, we explored the relationships between gut microbiome composition and environmental (location, time of year), life history (age, body mass, body condition), and health (disease status for EHD and BTV) metrics. Prior to our study, body condition was assessed and related to the survival of individual pronghorn [75]; we build on this previous work by investigating potential relationships between gut microbiome and body condition, with the goal of identifying a potential mechanism that may influence pronghorn survival. Based on previous literature regarding the relationships between microbiome and body condition [10–15], immunity (reviewed in [2, 3]), or environment [83, 86], we predicted pronghorn with differing body condition, serum disease status, or location would have different gut microbiome composition. Because limited work has been done on the pronghorn gut microbiome, our analyses were exploratory and descriptive in nature, with the goal of providing information for more targeted future research endeavors.

## Materials and methods

### Study area and capture methods

Our study used legacy samples and data collected from live-captured pronghorn in 2013 and 2014 from four study areas in south-central Wyoming: Baggs, Bitter Creek, Continental Divide-Creston Junction (hereafter CDC), and Red Desert (Fig 1). The focus of this earlier

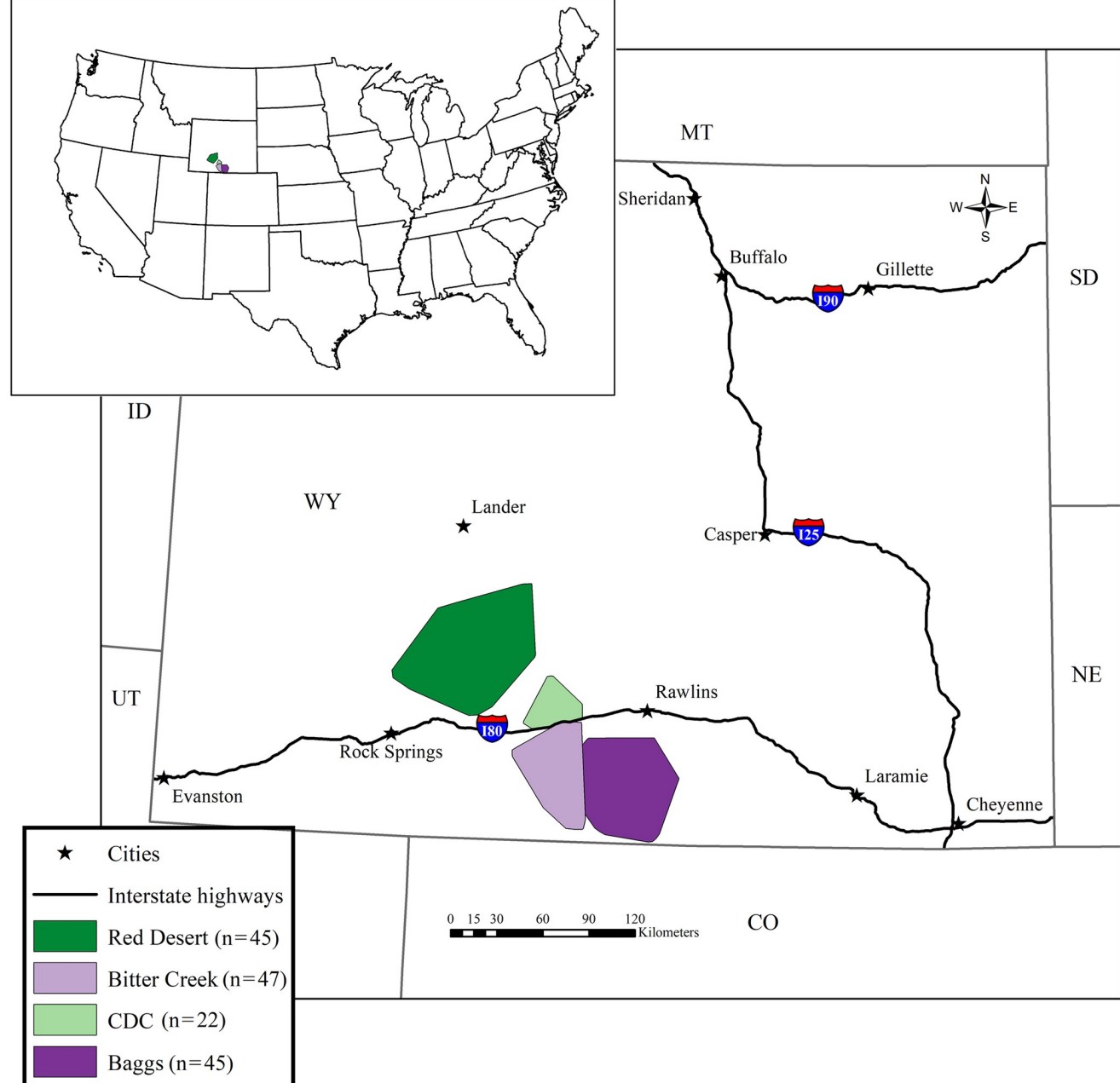

**Fig 1. Map of study area locations.** Location of the Red Desert, Continental Divide-Creston (CDC), Baggs, and Bitter Creek study areas with pronghorn numbers in each study area listed. Study area boundaries were delineated using a 100% minimum convex polygon including the pronghorn locations recorded within each study area.

study was to better understand how environmental and intrinsic factors and anthropogenic stressors affected pronghorn survival and seasonal habitat selection during daytime and nighttime by those pronghorn populations. The four study areas occurred within the Red Desert, an iconic landscape for pronghorn, where populations were declining in the face of environmental and anthropogenic changes including increasing energy infrastructure. Thus, the study areas from which we obtained legacy samples were selected to meet the objectives for this earlier research, which are further described in Reinking et al. 2018 and 2019 [67, 75]. The predominant vegetation community within these study areas was Wyoming big sagebrush (*A. tridentata wyomingensis*) with an herbaceous understory of perennial grasses and forbs. Low lying areas with alkaline or saline soils were dominated by black greasewood (*S. vermiculatus*) and Gardner's saltbush (*Atriplex gardneri*). In contrast, higher elevation areas were dominated by mountain big sagebrush (*A. t. vaseyana*), mixed shrub communities, and stands of aspen (*Populus tremuloides*). Other wildlife species common in the area included American badgers (*Taxidea taxus*), common raven (*Corvus corax*), elk (*Cervus canadensis*), greater sage-grouse, (*C. urophasianus*), mule deer (*O. hemionus*), sage thrasher (*Oreoscoptes montanus*), and white-tailed jackrabbit (*Lepus townsendii*). The topography of the area included sand deserts, rolling hills, badlands, and buttes. Oil and natural gas extraction, livestock grazing, and big game hunting were major land uses. For further information about the study areas, see [67, 75].

We captured 167 adult female pronghorn in November 2013 (n = 116), February 2014 (n = 13), and November 2014 (n = 38) [67, 75]. We used 159 samples in our analyses (n = 111 from November 2013, n = 13 from February 2014, and n = 35 from November 2014). For more information about the number of animals captured from each area in each capture period, see Table A in S1 Appendix. We captured pronghorn using helicopter net-gunning following the procedures of Jacques et al. [87] to reduce stress and capture-related mortality. During capture, we weighed animals to the nearest 0.1 kg and estimated age in half-year increments based on tooth eruption and wear [88]. A previous study corrected estimated age for this same group of animals based on cementum annuli analysis from dead animals [75]. Here, we applied the same correction factor to our age estimates to test whether this corrected age metric produced different results from estimated age. In addition, we measured the thickness of subcutaneous fat (mm) directly cranial to the cranial process of the tuber ischium [89] via ultrasound and assigned a leanness score based on the depth of the indentation (in inches) between the sacrosciatic ligament and caudal vertebrae (hereafter referred to as "ss-ligament") [75]. We collected fecal samples directly from animals by rectal palpation and froze samples in a chest freezer at -18˚C for later use. We took blood samples from the jugular vein using an 18-gauge, 2.54-cm needle, and the resulting samples were tested for two common diseases in ungulates: (1) Epizootic Hemorrhagic Disease (EHD) and (2) Bluetongue Virus (BTV). Blood samples were analyzed by the Wyoming Game and Fish Department Wildlife Health Laboratory/Wyoming State Veterinary Laboratory (Laramie, WY) to determine disease antibody status for each individual pronghorn for BTV or EHD (M. Miller, University of Wyoming, personal communication). We captured pronghorn a single time for this study, so data represents animal health at a single point in time. For more details on capture protocols, see [67, 75]. Pronghorn capture, handling, and monitoring procedures were approved by the Wyoming Game and Fish Department (Chapter 33–923 Permit) and the University of Wyoming Institutional Animal Care and Use Committee (protocol 20131028JB00037).

### Microbial community profiling

Pronghorn fecal samples were sequenced for gut microbiome composition using 16S rRNA amplicon sequencing [90], which involves amplifying the bacterial 16S ribosomal RNA

(rRNA) gene to determine which bacteria are present in the sample, and in what quantity. The DNA extraction, library preparation, and pooling were conducted by the Knight Lab in the Center for Microbiome Innovation at the University of California, San Diego (UCSD), and sequencing was conducted at the UCSD Institute for Genomic Medicine Genomics Facility using previously published methods [91] as described in [92]. Briefly, the Qiagen MagAttract PowerSoil DNA KF Kit was used for DNA extraction, in accordance with manufacturer protocols. The 515FB forward primer (5′ –GTGYCAGCMGCCGCGGTAA–3′) and the 806RB reverse primer (5′–GGACTACNVGGGTWTCTAAT–3′) were used to target the V4 region of the bacterial 16S rRNA gene. Samples were amplified in triplicates with 25μL polymerase chain reaction (PCR) reactions and then pooled samples before running on an agarose gel. Amplicons were quantified using a Quant-iT PicoGreen dsDNA Assay Kit following manufacturer instructions and cleaned using MoBio UltraClean PCR Clean-Up Kit following manufacturer instructions [91, 92]. The laboratory facility included eight extraction blanks for negative controls that contained no sample but were processed through their standard extraction and library preparation protocol. 16S libraries were sequenced on one lane of an Illumina MiSeq using 2 x 150 bp paired-end sequencing. Sequence data was stored under study number 12842 on the Qiita platform [93].

## Data analysis

We performed data processing and initial analysis using QIIME2 v. 2021.4 [94]. We de-multi-plexed FASTQ files, yielding 173 samples (including 8 blanks) and a total of 3,888,708 reads (mean read count ± SD = 22,478 ± 8,047 including blanks or 23,239 ± 7,209 excluding blanks). We performed de-noising and de-replication steps using the Divisive Amplicon Denoising Algorithm (DADA2) [95] module within the QIIME2 pipeline. After visual quality score inspection, we trimmed reads before base pair 12 and after base pair 150 to maintain high quality (Phred Score >30, for >95% of reads). After filtering, the total read count was 2,035,077 (mean read count ± SD = 12,114 ± 4,082 excluding blanks). Read counts at various steps of the DADA2 pipeline can be found in the supporting information (Table B in S1 Appendix). We found 3,389 unique amplicon sequence variant (ASV) sequences from this dataset. We used SILVA databases (Silva 138 SSURef NR99 515F/806R region sequences Silva 138 SSURef NR99 515F/806R region taxonomy and Silva 128 SEPP reference database) [96] to assign taxonomy to the species level, using QIIME2.

We conducted downstream analysis using R v. 4.3.0. We imported QIIME2 readable files (*.qza) into R using the package 'qiime2R' (v. 0.99.6) [97]. We used the 'phyloseq' package (v.1.44.0) [98] to remove non-bacterial (14 ASVs), mitochondrial (10 ASVs), and chloroplast (24 ASVs) ASVs, yielding 3341 ASVs. We used the package 'decontam' (v.1.20.0) [99] to decontaminate the remaining reads using the blanks as negative controls. We then removed negative controls and samples with ambiguous metadata, yielding 159 samples and 3,065 ASVs. We generated a rarefaction curve (S1 Fig) to determine a rarefaction cutoff of 4,936 reads for alpha diversity- or diversity within a sample- analysis, thereby excluding four samples due to low read count from alpha diversity analysis. During rarefaction, an additional 273 ASVs were removed. We calculated alpha diversity metrics including Simpson's diversity index, Shannon's diversity index, and observed richness using the 'phyloseq' package. For our analyses, we set the statistical significance at alpha = 0.10. Our alpha level was higher than the conventional level of 0.05 that is often used, however, we adjusted to a higher alpha to better align with our more exploratory research objectives [100]. We felt that with these objectives in mind, accepting a higher false positive (Type I) error rate would allow us lower false negative (Type II) error [101] and thus the ability to find more potential patterns that could be explored

in future studies. We compared alpha diversity metrics across the discrete pronghorn metrics (capture period, study area, and disease status) using Kruskal-Wallis tests, because assumptions of normality were not met for most comparisons. We made pairwise comparisons using a pairwise Wilcoxon test with a Bonferroni correction. To evaluate whether continuous life history metrics (age, body weight, body condition measures) were correlated with measures of alpha diversity, we performed Spearman's rank correlations between alpha diversity metrics and these continuous metadata metrics. Because some samples were missing capture data for certain metrics, we ran each analysis on the maximum number of samples possible of the 155 rarified samples that contained complete information for the chosen metric. As later analyses showed the presence of interactions, we conducted additional alpha diversity analyses within subsets of our groups. A description of these analyses and results can be found in S2 Appendix. We also ran linear regression models to look at the effect of the combined pronghorn metrics on observed richness. Details of this analysis can be found in S2 Appendix.

For analysis of beta diversity—or the differences in the diversity of species between ecosystems in a similar area—we transformed the read counts of our non-rarefied data to relative abundance. We divided the number of reads for each taxon within a sample by the total number of reads for that sample. We also log-transformed our data to test whether trends were any stronger when relative abundance was represented on a logarithmic scale to account for dominance of a few ASVs. However, patterns that we found mirrored results on the original scale (Tables H and I in S3 Appendix) and we chose to report our non-log transformed data for more intuitive interpretation. For more information on log-transformation and analysis see S3 Appendix. To visualize gut microbiome differences among pronghorn for each metric, we generated ordination plots using Principal Coordinate Analysis (PCoA) using the Bray Curtis dissimilarity metric and included all 159 samples. In addition, we used the envfit() function in the 'vegan' package (v. 2.6.4) [102], to investigate how continuous values of age, weight, and ss-ligament were related to explanatory PCoA axes. We performed permutational multivariate ANOVA (PERMANOVA) tests with 999 permutations using the adonis2 function [103] in the 'vegan' package, also using Bray Curtis as a distance metric.

As our study was somewhat exploratory in nature, we completed a two-phase analysis to better understand important patterns. We first performed PERMANOVA tests on pronghorn metadata metrics individually to test for differences ($p < 0.100$), with continuous variables binned to allow for comparisons among groups. In the second phase, for each metric of interest, we chose one of the variables representing each pronghorn metric for input into a model, avoiding inputting multiple factors that represented the same metadata metric. Our combined PERMANOVA investigated the marginal effects of the different metrics and included study area; BTV status; EHD status; weight by 5-kg increments; age by young, middle age, and old groupings; and ss-ligament. We also performed the combined PERMANOVA with interactions between study area and the pronghorn intrinsic measures to investigate if the relationships between pronghorn gut microbiome and pronghorn age, disease status, body fat, weight, or age varied in different study locations. For more information on initial exploratory analysis, binning, and choice of factors for our second phase of analysis see S3 Appendix.

To ease the interpretation of the effects of the continuous pronghorn metrics of age, weight, and ss-ligament, we also performed a Redundancy Analysis (RDA). We centered and standardized the variables of age, weight, and ss-ligament using the decostand() function in the 'vegan' [102] package. We transformed the relative abundance of the microbial community using the Hellinger transformation to avoid issues of double zeros in community abundance data increasing the similarity between sites [104]. We performed RDA analysis both with the reduced set of continuous metrics of age, weight, and ss-ligament as well as the full set of metrics including discrete variables of study area, EHD status, and BTV status. After performing

the RDA, we calculated the adjusted $R^2$ to correct for our number of explanatory variables using the RsquareAdj() function in the 'vegan' package [102] and determine the amount of variance explained.

## Results

### Pronghorn life history, environmental, and health metrics

Our final data set included information from 159 pronghorn. Of these, 30 animals (18.87%) were positive for EHD and 21 animals (13.21%) were positive for BTV, including 10 animals (6.29%) positive for both diseases. We included 45 animals in Baggs, 45 in Red Desert, 22 in CDC, and 47 in Bitter Creek study areas. Pronghorn ages ranged from 1 to 12 years, while our corrected pronghorn ages ranged from 2.80 to 11.44 years. Pronghorn weights ranged from 39.66 to 59.46 kg. The measures for indentation of ss-ligament ranged from 0–3.81 cm (0–1.50 inches), with higher values indicating a greater depression above the sacrosciatic ligament (i.e., less body fat padding this area). Measurements for maximum rump fat thickness ranged from 0 to 7 mm. For more information about pronghorn metric values within each study area see Table C in S1 Appendix.

### Microbiome composition

Bacterial composition of pronghorn fecal samples included 23 phyla, 35 classes, 83 orders, 143 families, 267 genera, and 308 species (Table 1). Composition of the bacterial community was dominated by the phyla Firmicutes (now Bacillota), followed by Bacteroidota across all study areas (Fig 2). Even at finer taxonomic levels (e.g., Family), 15 core bacterial groups dominated the community and represented approximately 95% of our assigned reads (Table 1). The composition of bacterial phyla, classes, orders, and families was similar across samples from different study areas (Figs 2–5). Common families included Bacteroidaceae, Christensenellaceae, Lachnospiraceae, Oscillospiraceae, and Ruminococcaceae (Fig 5). We observed this similar pattern of dominant bacterial groups when we grouped samples by other metrics (S2–S4 Figs), albeit with variation among individuals (S5 Fig).

### Alpha diversity

We found few statistical differences when assessing alpha diversity relative to pronghorn metrics. For our discrete metrics, we found differences ($p < 0.100$) in all alpha diversity metrics between the different capture periods (Table 2) with November 2013 samples having higher

**Table 1. Taxonomic depth and breadth of read assignments.**

| | Total Members Identified | Percent reads assigned to depth | Top 10 (% of assigned reads) | Top 15 (% of assigned reads) | Top 20 (% of assigned reads) |
|---|---|---|---|---|---|
| Phylum | 23 | 99.999% | 99.995% | 99.999% | 100% (rounded) |
| Class | 35 | 99.994% | 99.906% | 99.980% | 99.998% |
| Order | 83 | 98.245% | 97.781% | 99.251% | 99.723% |
| Family | 143 | 98.060% | 87.324% | 95.295% | 97.431% |
| Genus | 267 | 85.948% | 75.053% | 84.484% | 88.366% |
| Species | 308 | 54.486% | 61.415% | 71.381% | 78.034% |

The first two columns indicate the number of members identified at each taxonomic level as well as the percentage of total sample reads that were assigned to that taxonomic level. The following columns show the percentage of read counts composed by the top 10, 15, or 20 members of the community at a given taxonomic level, showing the presence of a core group of microbes. Note: when taxonomy was assigned not all amplicon sequence variants (ASVs) could be assigned down to each taxonomic level, so percentages in the final three columns are representative of the reads that could be assigned to that level.

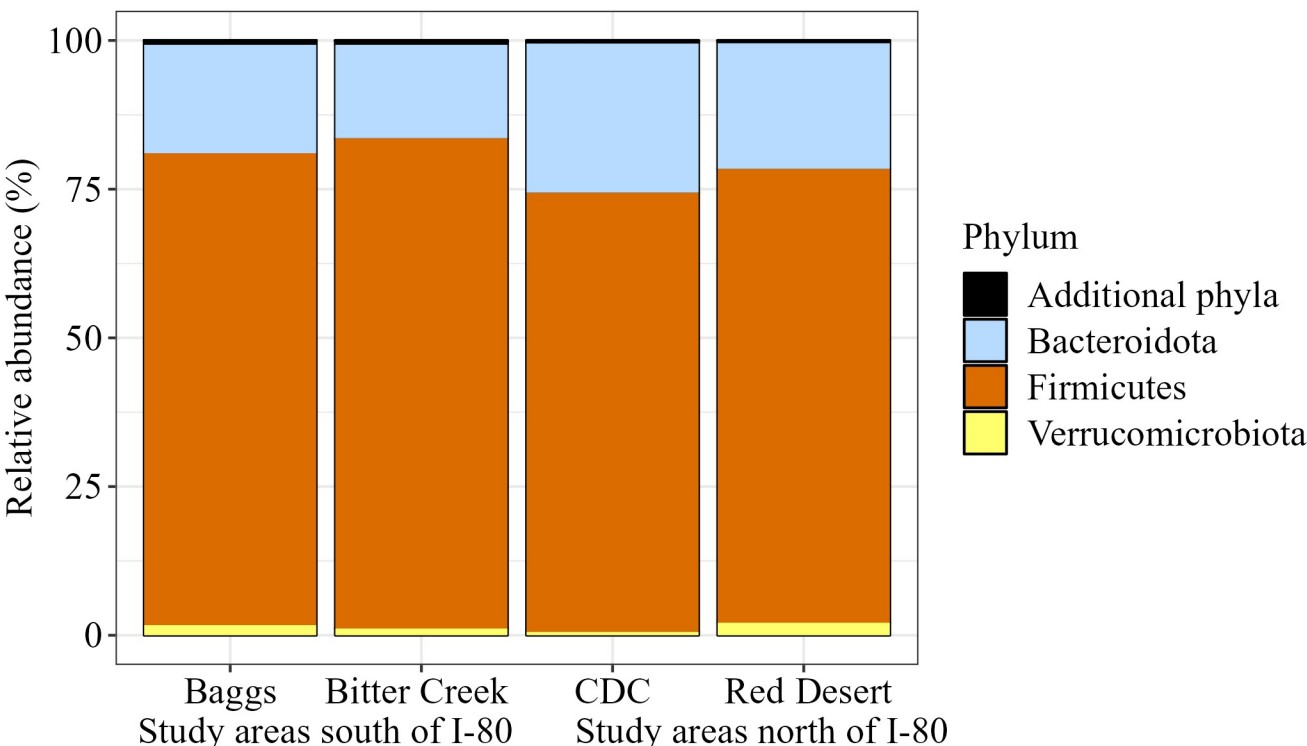

**Fig 2. Pronghorn microbiome composition represented by phylum relative abundance.** Phyla relative abundance is grouped and visualized by study area and location relative to Interstate 80 (I-80). The top 3 phyla depicted represent 99.382% of the assigned amplicon sequence variants (ASVs) present in our samples.

Shannon and Simpson indices compared to both November 2014 and February 2014 samples, and November 2013 and 2014 samples having higher observed richness than February 2014 samples (Table 3, Fig 6A). The study area did not affect this general pattern observed between capture periods (Fig 6B). For animals that were either north or south of Interstate 80 (I-80), we found Shannon's diversity index (mean ± SE: north = 4.334 ± 0.034, south = 4.412 ± 0.032) and observed richness measures (north = 186.281 ± 5.653, south = 201.099 ± 5.017) were different (Table 2), with animals south of the highway having higher alpha diversity (Table 3, Fig 6C). Simpson's diversity index did not differ between pronghorn north or south of I-80, while no diversity indices differed between animals in different study areas or with differing disease statuses (Table 2). We did not observe strong correlations with alpha diversity metrics and any of our continuous life history metrics of age, weight, and body condition (Table 4). The models used to evaluate the combined effects of pronghorn metrics on alpha diversity supported the results we found when looking at metrics singularly (S2 Appendix). Pronghorn north and south of I-80 showed overall similar relationships between alpha diversity metrics and pronghorn metrics with some slight differences (S2 Appendix).

## Beta diversity

We found that the first two axes of PCoA using the Bray Curtis distance for ordinations accounted for 13.6% (7.6% and 6.0% respectively) of the variation among our samples (Fig 7). When we visualized with PCoA ordination plots, microbial communities appeared to group clearly when partitioned by study area (Fig 7A) in addition to whether a study area was north

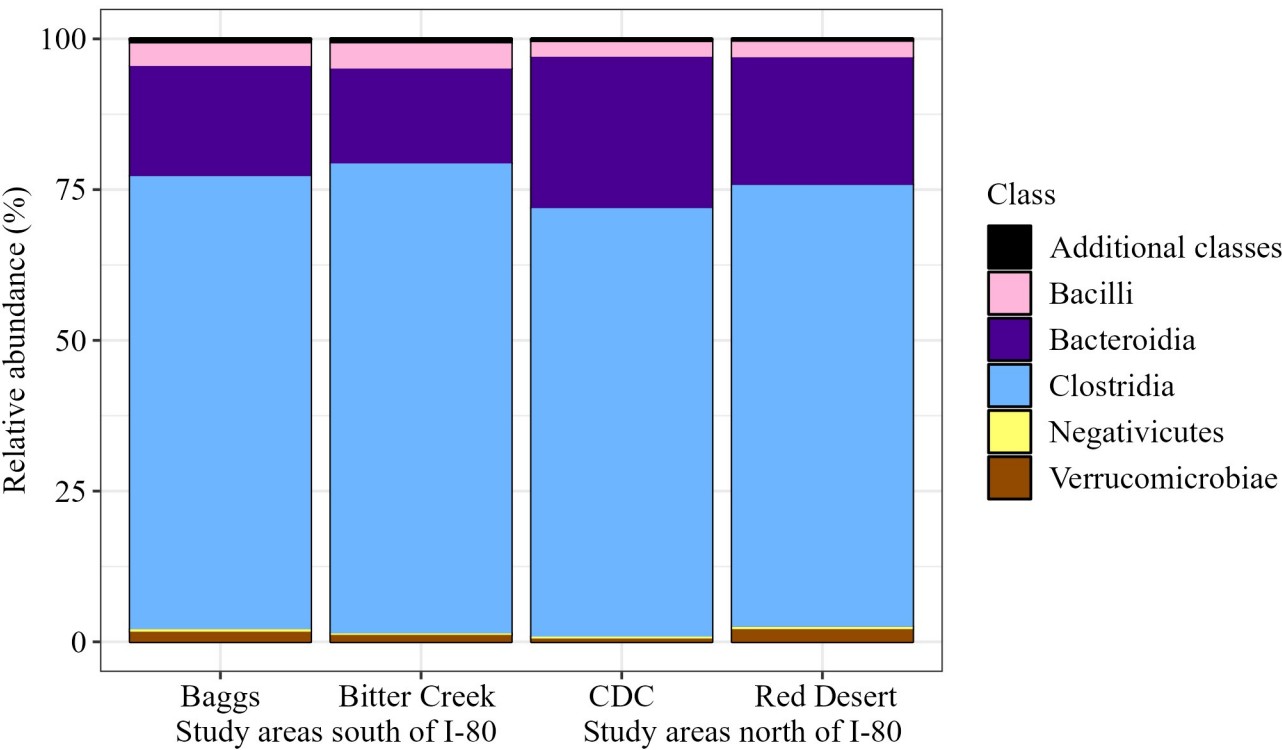

**Fig 3. Pronghorn microbiome composition represented by class relative abundance.** Class relative abundance is grouped and visualized by study area and location relative to Interstate 80. The top 5 classes depicted represent 99.355% of the assigned amplicon sequence variants (ASVs) present in our samples.

or south of I-80 (Fig 7A; note shape beta dispersion). None of the continuous life history metrics of age, weight, or ss-ligament were related to our PCoA axes 1 and 2 (Fig 7B). While we observed some separation between the November 2014 group compared to November 2013 and February 2014 groups (Fig 7C), it is difficult to attribute this solely to a capture period effect, as the November 2014 group was dominated by individuals from the CDC study area, and there was only a single capture period at this location (Table A in S1 Appendix).

Our first phase of PERMANOVA analysis indicated differences based on the study area, location north or south of I-80, capture period and season, EHD status, and ss-ligament (Table H in S3 Appendix) when run as single metrics. This first stage of analysis also indicated that the best explanatory variable for location was study area versus whether the area was north or south of I-80, although both were highly significant (p = 0.001; Table H in S3 Appendix). For more information on the results of the first phase of PERMANOVA and our choice of metrics for our second phase of analysis see S3 Appendix.

In the second phase of analysis, where we combined the study area, ss-ligament, categorical age, binned weights, BTV, and EHD metrics together into a PERMANOVA testing for marginal effect of each variable, we found significant differences only for study area (p = 0.001; Table 5) with both EHD (p = 0.823) and ss-ligament (p = 0.253) no longer important when the marginal effects of other variables were accounted for (Table 5). When we added interactions between study area and each pronghorn intrinsic metric, we found the interaction between study area and ss-ligament (p = 0.035) and study area and age (p = 0.077) to be important (p < 0.100), while interactions between study area and either BTV (p = 0.691), EHD

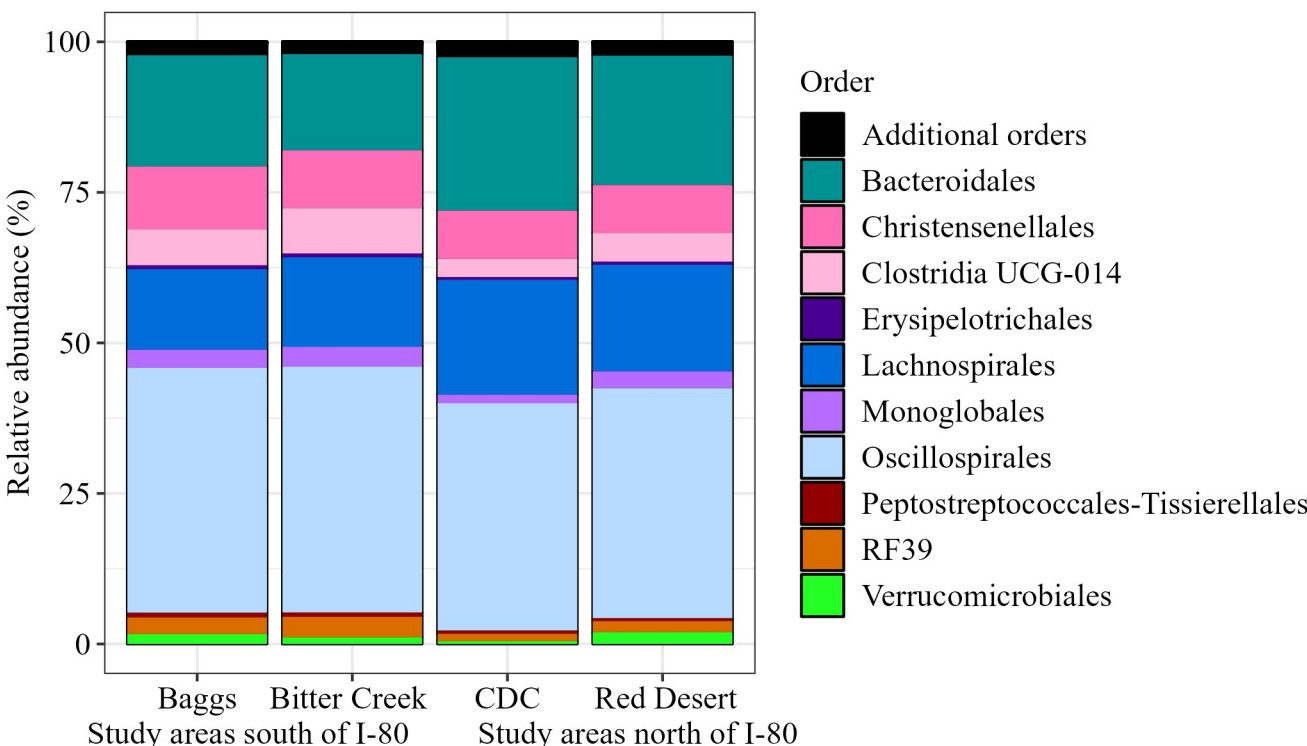

**Fig 4. Pronghorn microbiome composition represented by order relative abundance.** Order relative abundance is grouped and visualized by study area and location relative to Interstate 80 (I-80). The top 10 orders depicted represent 97.781% of the assigned amplicon sequence variants (ASVs) present in our samples (Table 1).

(p = 0.643), or weight (p = 0.800) were not significant (Table 5). This indicated that the effect of age or ss-ligament may differ across study areas, so we subset our data by study area to explore this. Further PERMANOVAs within subsets of single study areas showed that within the Red Desert study area, there were differences in the microbiome of pronghorn of different ages (p = 0.029, Table M in S4 Appendix). However, we found no differences related to age within other study areas (Tables J-L in S4 Appendix). Ss-ligament was not statistically related to the microbiome composition within any subset of a single study area (Tables J-M in S4 Appendix). Results of analyses on subsets of single study areas are detailed in S4 Appendix.

When we completed RDA modeling including only continuous variables of age, weight, and ss-ligament, we did not produce a significant model (p = 0.889). However, when we added study area, EHD status, and BTV status, a significant model (p = 0.001) resulted with the variables for study area (p = 0.001) and EHD status (p = 0.050) being important. (p < 0.100). However, the effects were subtle with only 8.8% (3.1% adjusted) of the microbial community composition explained by our included pronghorn life metrics. Overall, the results from our RDA analysis support the results from the PERMANOVA, with the study area showing a significant effect but only explaining a small portion of the variation in the gut microbiome community.

## Discussion

To our knowledge, this is the first study of the bacterial gut microbiome of pronghorn. Thus, our first objective was to describe the gut microbial composition. We found that the core

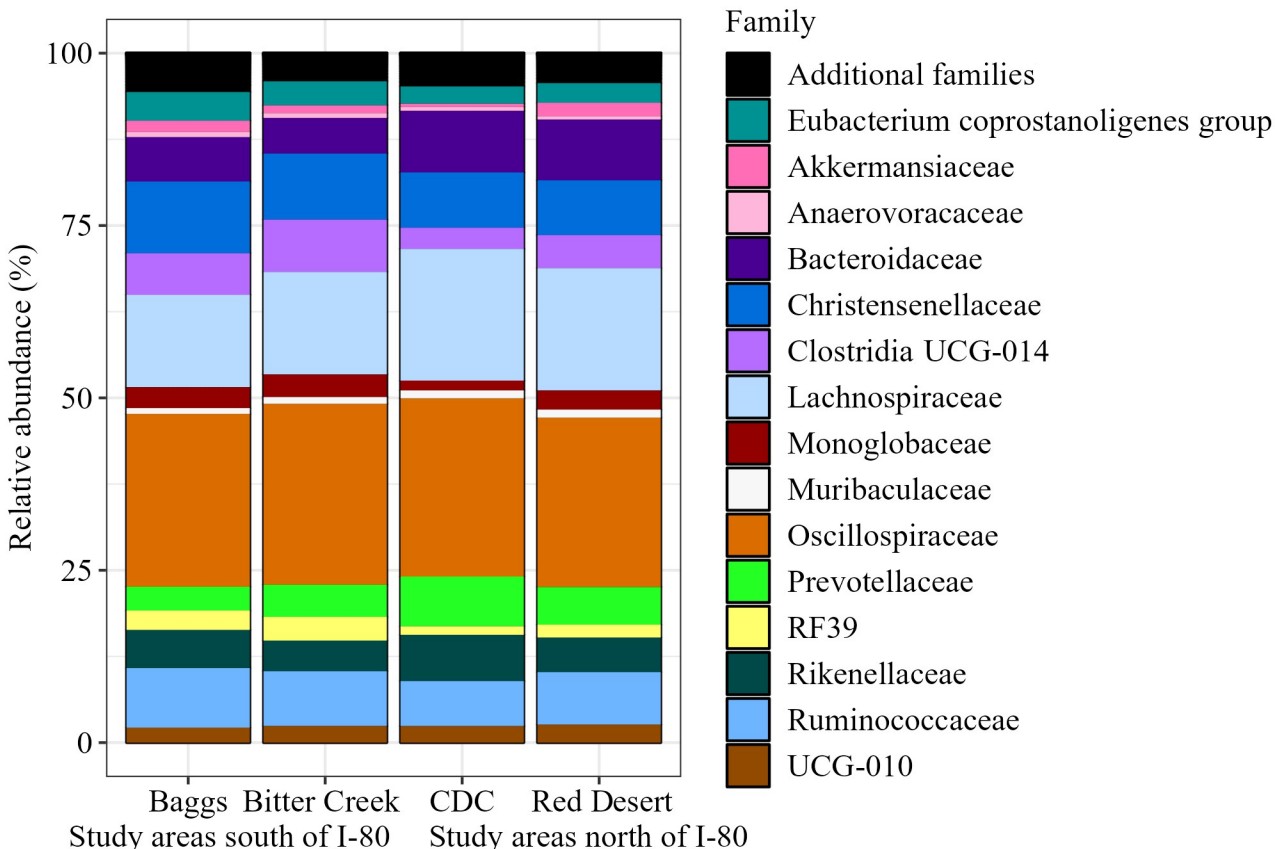

**Fig 5. Pronghorn microbiome composition represented by family relative abundance.** Family relative abundance is grouped and visualized by study area and location relative to Interstate 80. The top 15 families depicted represent 95.295% of the assigned amplicon sequence variants (ASVs) present (Table 1).

microbiome—or the set of microbes that are characteristic of a specific environment or host species—exhibited high consistency in pronghorn in the Red Desert. The phyla Firmicutes (now Bacillota) and Bacteroidota dominated pronghorn gut microbiomes across all study areas. Even at finer taxonomic scales (e.g., order, class, family, and genus), we saw a similar

**Table 2. Statistical comparison of three alpha diversity metrics among discrete pronghorn metrics.**

| | n | Shannon | | Simpson | | Observed Richness | |
|---|---|---|---|---|---|---|---|
| | | $\chi^2$ | p | $\chi^2$ | p | $\chi^2$ | p |
| Study Area | 155 | 3.204 | 0.361 | 2.150 | 0.542 | 4.433 | 0.218 |
| BTV | 152 | 0.587 | 0.444 | 0.189 | 0.663 | 0.085 | 0.771 |
| EHD | 152 | 2.059 | 0.151 | 0.214 | 0.643 | 1.968 | 0.161 |
| **I-80** | 155 | **2.868** | **0.090** | 0.611 | 0.435 | **3.648** | **0.056** |
| **Capture period** | 155 | **20.698** | **<0.001** | **16.391** | **< 0.001** | **12.441** | **0.002** |

Comparisons of Shannon's diversity index, Simpson's diversity index, and observed richness in the microbial community across pronghorn metrics including study area, BTV and EHD status, location relative to I-80, and capture period. Observed richness represents amplicon sequence variant (ASV) richness. We report maximum sample sizes (n), available for each metric. To maintain consistency in comparisons, we conducted a non-parametric Kruskal-Wallis test ($\chi^2$) as normality assumptions were not met for all metrics. **Significant differences are bolded (p < 0.100).**

**Table 3. Values for alpha diversity metrics for groups of pronghorn.**

|  | n | Shannon | Simpson | Observed Richness |
|---|---|---|---|---|
| **Capture period** | 155 |  |  |  |
| November 2013 | 110 | 4.448 ± 0.025 (a) | 0.971 ± 0.001 (a) | 201.882 ± 4.492 (a) |
| February 2014 | 11 | 4.122 ± 0.079 (b) | 0.964 ± 0.003 (b) | 154.000 ± 10.308 (b) |
| November 2014 | 34 | 4.241 ± 0.051 (b) | 0.959 ± 0.003 (b) | 185.912 ± 7.499 (a) |
| **I-80** | 155 |  |  |  |
| North | 64 | 4.334± 0.034 (a) | 0.968 ± 0.002 (a) | 186.281 ± 5.653 (a) |
| South | 91 | 4.412 ± 0.032 (b) | 0.968 ± 0.002 (a) | 201.099 ± 5.017 (b) |

Values for the above alpha diversity metrics within pronghorn metrics that showed significant differences in Kruskal-Wallis tests. Values reported for mean (± SE) during each capture period and in areas north or south of I-80. Matching letters denote when values for different capture periods or locations relative to I-80 are not significantly different (p > 0.1) using a Wilcoxon rank sum test with Bonferroni correction.

group of 10–15 bacterial ASVs constituting the majority (>95%) of the gut microbiome in these animals, with the composition similar when animals were grouped by various metrics. These results agree with other gut microbiome studies of North American wild and domestic herbivores, which demonstrate Firmicutes and Bacteroidota make up a large proportion of the gut microbiome [7, 8, 83, 105]. Previous studies have shown both of these phyla to be important in breaking down complex carbohydrates [106]. Bacteroidota tend to be more important for polysaccharide breakdown [107] as evidenced by their evolution of polysaccharide utilization loci [108], while various Firmicutes have shown to be integral to cellulose degradation [109]. As previous literature on the pronghorn bacterial gut microbiome is lacking, and pronghorn are not listed within the existing Animal Microbiome Database [110], our analysis is a foundational descriptor of the pronghorn gut microbiome as represented by fecal samples, providing a novel starting point for future research endeavors. In relation to our second objective, we found relationships between pronghorn gut microbiome composition and both study area and capture period, while relationships between gut microbiome composition and life history and health metrics had more subtle effects.

Similarities in gut microbial communities of animals within the same geographic area and differences between animals in different geographic areas may be attributed to interactions with conspecifics, similar habitats (plants available or environmental conditions), or both. Our finding of beta diversity differences between study areas is supported by both our PERMANOVA and RDA results and agrees with previous studies in wild mammals. For example, differences have been found in the gut microbiomes of mule deer in different seasons and geographic locations [83] and brown bears (*Ursus arctos*) living under different environmental conditions [86]. Additionally, studies in equines [111], baboons (*Papio cynocephalus*) [112], and humans [113] have shown that an animal's gut and other microbiomes can be affected by social interactions (review in [114]). We also observed differences in alpha and beta diversity when our study animals were grouped into those individuals occurring north or south of I-80 (Table 2, Fig 7A, and Table H in S3 Appendix). Highways with higher traffic levels or non-wildlife-friendly fencing can become complete barriers to pronghorn [64], and I-80 has been identified as a largely impermeable barrier to pronghorn movement [115]. The inability of pronghorn to easily cross I-80 and acquire microbes from novel environments or conspecifics via social interaction may contribute to these different gut microbiome communities (i.e., microbiome dispersal limitation). Moeller et al. [116] found that the diversification of gut microbiome composition of various mammals was affected by the barrier effect that

## A: Richness by Capture Period

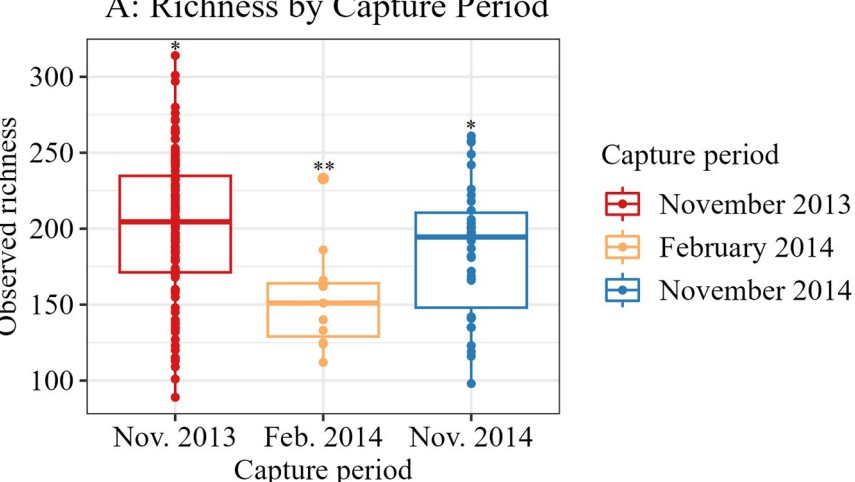

## B: Richness by Capture Period and Study Area

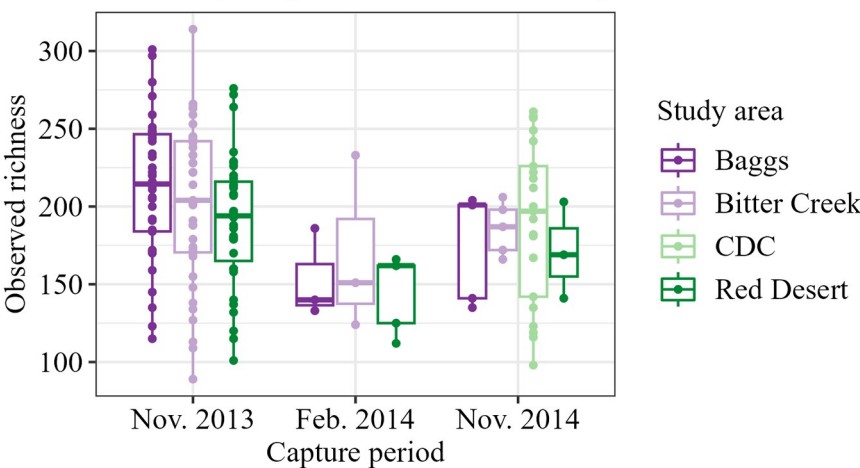

## C: Richness by Location Relative to I-80

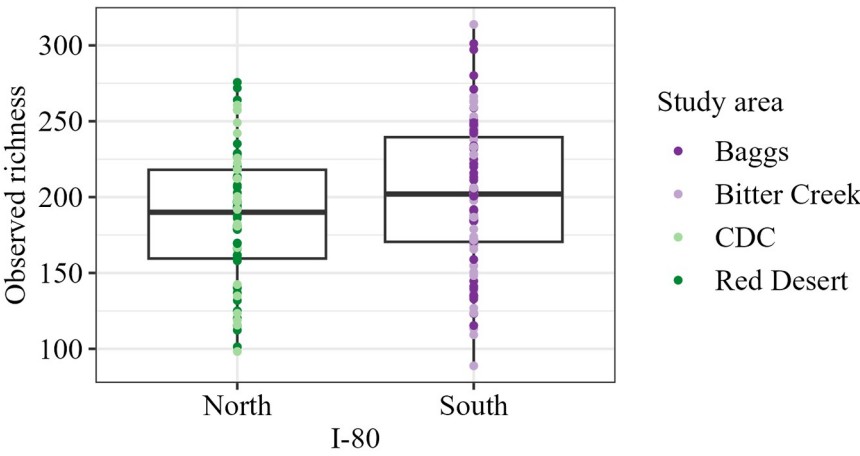

**Fig 6. Visualization of alpha diversity differences amongst pronghorn study metrics.** A) Observed richness for pronghorn captured during 3 capture periods (p = 0.002) with animals captured in February 2014 having lower observed richness than animals captured during both November periods. Differing numbers of * indicate significant differences (p < 0.100). B) The same visualization, now showing that trend of lower alpha diversity in February 2014 was consistent across study areas. C) Animals south of Interstate 80 (I-80) exhibited higher observed richness than animals north of I-80 (p = 0.070), study areas within are depicted by different colors.

**Table 4. Spearman's rank correlations ($r_s$) between alpha diversity measures and continuous pronghorn life history metrics.**

| Pronghorn Metric | n | | Alpha Diversity Metric | | |
|---|---|---|---|---|---|
| | | | Observed Richness | Shannon Diversity | Simpson Diversity |
| Age | 152 | $r_s$ | -0.027 | 0.015 | 0.045 |
| | | p | 0.741 | 0.854 | 0.578 |
| Corrected age | 152 | $r_s$ | -0.027 | 0.015 | 0.045 |
| | | p | 0.741 | 0.854 | 0.578 |
| **Body weight (kg)** | 150 | $r_s$ | -0.080 | -0.135 | -0.138 |
| | | p | 0.329 | **0.010** | **0.093** |
| Ss-ligament | 144 | $r_s$ | -0.013 | 0.038 | -0.007 |
| | | p | 0.875 | 0.654 | 0.930 |
| Max fat | 144 | $r_s$ | -0.064 | -0.131 | -0.087 |
| | | p | 0.447 | 0.116 | 0.298 |

Alpha diversity metrics of Shannon diversity index, Simpson diversity index, and observed richness and their correlations to age, corrected age, body weight, and body condition metrics. Note: although correlation of body weight and both Shannon and Simpson diversity index was significant at the alpha = 0.100 level the accompanying correlations were not strong. **Significant correlations are bolded (p < 0.100).** We conducted correlations on the 155 pronghorn included in the rarified dataset, ignoring missing observations for each metric of interest. Thus, sample size (n) reports the number of observations included for each correlation.

geographical distance presents to bacterial dispersal. In our case, I-80 could be acting as a similar barrier to microbial dispersal in pronghorn populations.

Pronghorn gut microbiome composition differed by capture period in many of our tests. While our sample representation from different seasons was not consistent due to the uneven sampling nature of the legacy study, capture period and thus season showed differences in gut microbiome composition throughout many of our analyses and warrants further investigation in future studies. Alpha diversity, as measured by Shannon's index, Simpson's index, and observed richness, was lower during February 2014 than either one or both November capture periods, depending on the chosen diversity measure of interest (Table 3, Fig 6A). This pattern was consistent across study areas, with observed richness trending lower in February compared to November in the Baggs, Bitter Creek, and Red Desert study areas (Fig 6B). Beta diversity also differed by capture period in our single factor PERMAONVAS (Table H is S3 Appendix), with November 2014 samples appearing distinct from the other two groups in ordination (Fig 7C). However, because all animals in the CDC study area were captured in November 2014 (Table A in S1 Appendix), we cannot attribute this effect solely to the capture period. Our finding of potential seasonal effects on the gut microbiome is consistent with findings in mule deer in Utah, USA, which were shown to have different gut microbiomes in December as compared to March [83]. Although these researchers were unable to establish causal links, they suggested potential mechanisms that could explain relationships between select microbial taxa and winter physiologic changes of mule deer, such as fat or protein catabolism [83]. Pronghorn also experience similar changes across seasons including metabolization of fat stores [117], shifts in diet [49, 51, 52], or seasonal behavior changes such as changes in group size [48, 118]. Any of these seasonal shifts in pronghorn life history have the potential to influence gut microbiome composition, a possible explanation for the differences we observed in alpha and beta diversity between capture periods. Unfortunately, we were unable to include capture period, and therefore the effect of season, in our combined PERMANOVA and RDA models as some pronghorn metrics of interest were not recorded during February captures. The effect of season should be investigated in future studies of wild animal gut microbiomes to better understand how microbial composition changes across seasons in different species.

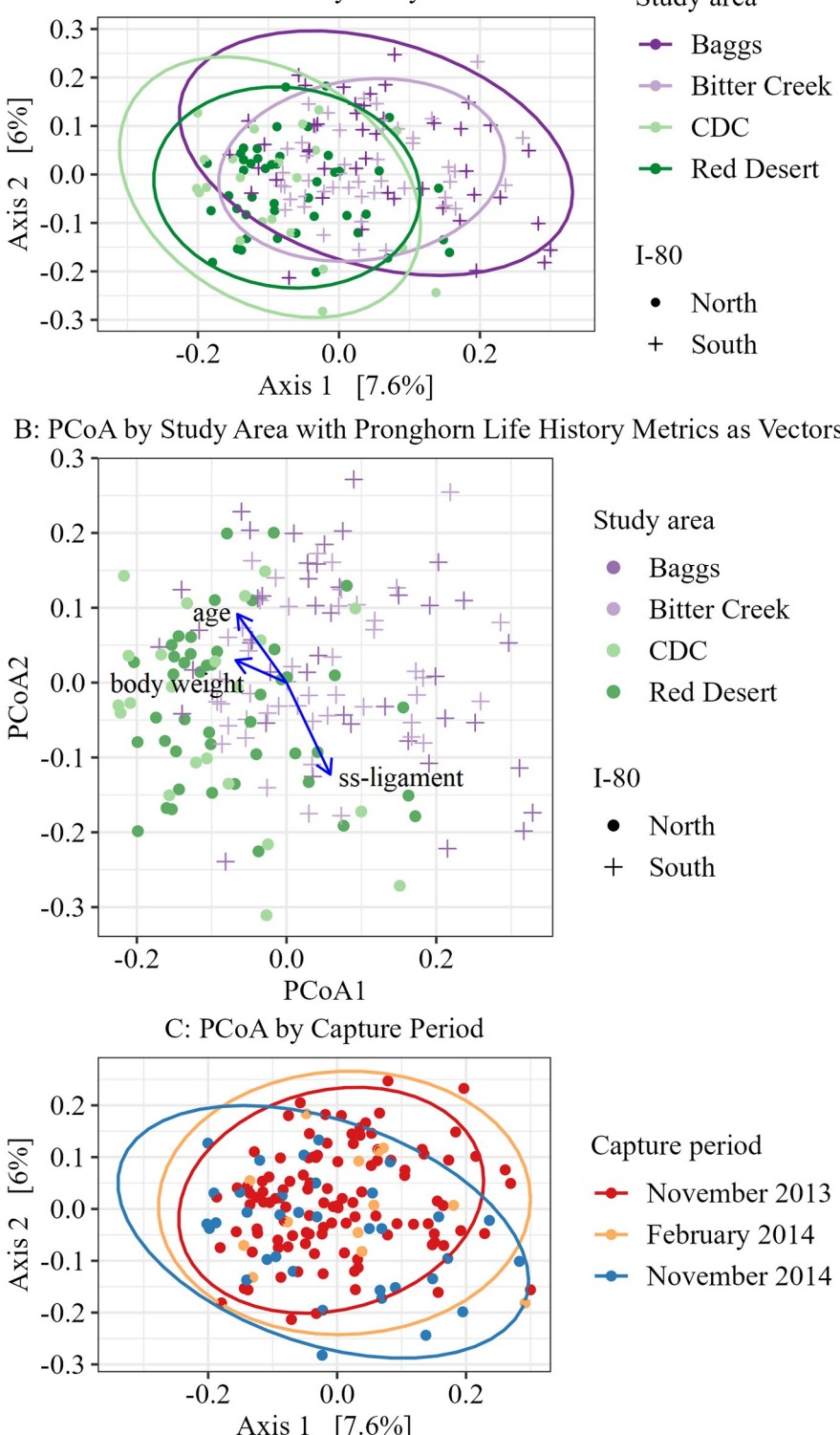

**Fig 7. PCoA ordinations using Bray-Curtis dissimilarity measure.** Data represent counts transformed to relative abundance without rare taxa removed and are grouped by (A) study area with ellipses of 95% CI, (B) study area with vectors representing continuous variables of explanatory pronghorn life history metrics of age, ss-ligament, and weight, and (C) capture period with ellipses of 95% CI. The first two axes explain 13.6% of the variation in the samples. Note in 7A that Baggs and Bitter Creek (south of Interstate 80, plus signs) are more similar and Continental Divide-

Creston (CDC) and Red Desert (north of Interstate 80, circles) are more similar. 7B shows the addition of vectors representing age, weight, and ss-ligament, however these were not significantly related to PCoA axis 1 or 2.

Identifying which microbial taxa are involved in seasonal shifts may prove important to understanding the mechanisms for how host animals cope with nutrient scarcity in the winter or could inform research into selecting specific microbial taxa for use in probiotic applications or as animal health bio-indicators.

In addition to geography and season, we investigated relationships between gut microbiome composition and animal health markers such as body condition and disease status. Previous research has demonstrated links between body condition and gut microbiome composition including relationships with feed efficiency in livestock [7–9] and links between gut microbiome composition and fat deposition, body condition, and metabolism in both mice and humans [10–15]. Therefore, we predicted we would see differences in the gut microbiome of pronghorn with differing levels of body condition. Ss-ligament, a measure of body condition, was previously related to pronghorn survival in our study location [75]. Therefore, we chose this specific body condition metric in our analyses because a link between ss-ligament and gut microbial composition could show potential for a link between gut microbiome and survival in pronghorn. When testing metrics with individual PERMANOVAs, differences in gut microbiomes of animals with varying ss-ligament measures seemed to be supported (Table H in S3 Appendix). However, our combined PERMANOVA model suggests body

**Table 5. Model fit statistics from PERMANOVA ran with multiple metrics and multiple metrics with interactions.**

| Model before interactions | | | | | |
|---|---|---|---|---|---|
| Term | F | df | $R^2$ | % Variation attributed | P |
| **Study area** | **2.752** | **3** | **0.059** | **5.9%** | **0.001** |
| EHD | 0.845 | 1 | 0.006 | 0.6% | 0.823 |
| SS-ligament | 1.036 | 10 | 0.074 | 7.4% | 0.253 |
| BTV | 1.004 | 1 | 0.007 | 0.7% | 0.423 |
| Age (young, middle, old) | 1.121 | 2 | 0.016 | 1.6% | 0.173 |
| Weight (5 kg increments) | 0.931 | 3 | 0.020 | 2.0% | 0.761 |
| Model with interactions | | | | | |
| Term | F | df | $R^2$ | % Variation attributed | P |
| Study area | 3.097 | 3 | 0.066 | 6.6% | 0.001 |
| EHD | 1.166 | 1 | 0.008 | 0.8% | 0.150 |
| SS-ligament | 1.057 | 10 | 0.075 | 7.5% | 0.191 |
| BTV | 1.102 | 1 | 0.008 | 0.8% | 0.225 |
| Age (young, middle, old) | 1.173 | 2 | 0.017 | 1.7% | 0.097 |
| Weight (5 kg increments) | 0.943 | 3 | 0.020 | 2.0% | 0.698 |
| Study Area x EHD | 0.955 | 3 | 0.020 | 2.0% | 0.643 |
| **Study Area x SS-ligament** | **1.088** | **18** | **0.138** | **13.8%** | **0.035** |
| Study Area x BTV | 0.942 | 3 | 0.020 | 2.0% | 0.691 |
| **Study Area x Age (young, middle, old)** | **1.106** | **6** | **0.047** | **4.7%** | **0.077** |
| Study Area x Weight (5 kg increments) | 0.950 | 9 | 0.060 | 6.0% | 0.800 |

Test statistic (F), degrees of freedom (df), $R^2$, and p are reported for each metric. Factors significant at the p <0.100 level are bolded. Sample size for combined PERMANOVAs was 134. When interactions were not present, the PERMANOVA was run to account for the marginal effect of each variable. When the model was run with interactions there were significant interactions between study area and ss-ligament and study area and age.

condition, as indexed by ss-ligament, to have a more context-dependent relationship with gut microbiome composition, with the effect of ss-ligament not significant on its own in the combined model, but having a significant interaction with study area when interactions were added to the combined model (Table 5). In addition, measures of alpha diversity were not correlated with values for body condition, except for a subtle relationship within a subset of only animals south of I-80 (Table F in S2 Appendix). Therefore, while some of our analyses provide support for subtle gut microbiome differences in animals of varying body condition, our results do not conclusively show a clear link between gut microbiome composition and measures for body condition. However, the potentially complex trends we saw warrant further investigation into the relationship between gut microbiome and body condition. We saw some support for differences in animals with differing EHD status in our single metric PERMANOVAs and RDA, but we did not see differences in gut microbiome community related to disease status for BTV or EHD in our final PERMANOVA model. In addition, within our alpha diversity analyses we only saw alpha diversity differing by disease status in a subset of animals north of I-80 (Table D in S2 Appendix). These results suggest that the pronghorn gut microbiome in our study may be resilient across these disease exposures. Similar results regarding disease were found in gut microbiomes of moose in Minnesota, with pathogen exposure not predictive of gut microbial community. As in our study, pathogens in these moose were detected by serological evidence, showing evidence of previous exposure rather than current infection [105]. Similar to the moose study, we reason that the lack of consistent disease effect on the gut microbiome could be due to animals not having a current infection, or that the animal's gut microbiomes are resilient to pathogen exposure.

While we did find that gut microbial communities differed based on some of our study metrics, these metrics only explained a small amount of the variation in the pronghorn gut microbiome. This may be due to the consistency in the core bacteria that dominated the pronghorn gut microbiome. This consistency in gut microbiome with only subtle differences may be seen because we only evaluated a single species of host animal. A study in seven species of woodrats (*Neotoma* spp.) found that while diet, host phylogeny, and geography collectively explained 49% of the variation in the wild woodrat gut microbiome composition, host species had the greatest effect, explaining 35% of the variation [119]. Another possibility is that individual variation, known to explain large amounts of variation in the gut microbiome in other species, was not included in our study. A study of horses that collected multiple samples over time from the same animals found large variation among individuals, and individual animal ID accounted for about 50% of the variation in the samples [111]. Due to concerns about capture-related mortality during repeat captures [75], we only implemented a single capture for each individual, so we were unable to investigate this effect. However, when assessing the gut microbial community of individual animals (S5 Fig), we observed that some individuals possessed gut microbiome compositions divergent from the general patterns. It is possible that within pronghorn, gut microbial differences are largely unique for individuals, and thus, we saw only subtle differences when comparing grouped animals. In order to properly account for this effect, we would need a study design that collects repeated samples from the same individual pronghorn. Diet has also been shown to have a large influence on gut microbial composition. This is evident when comparing animals with diverse diets and gut types [21], and in morphologically similar animals. A study of two similar woodrat species (*N. bryanti and N. lepida*) consuming different plants found that gut microbiome composition was associated with the common diet plants of each species [120]. Another study of woodrats found that diet richness was correlated with gut microbiome richness, and diet explained 16% of the variation in microbial composition [119]. We do not have data regarding the diet of pronghorn in our study, however, it is possible that because all captures occurred during the dormant season for plants, pronghorn

may only have had limited forage availability, thus similar diets, leading to similarities in gut microbiomes. Therefore, since our study includes hosts of the same species, and we could not include the effect of diet or individuality—all factors that have been shown to be highly explanatory of gut microbiome composition—it is logical for our results to only show subtle effects.

## Conclusions

The microbial community can change and adapt within an animals' lifetime; thus, microbiome changes could enable hosts to adapt more quickly to changing conditions [121]. It has been proposed that plasticity in the gut microbiome may lead to greater host adaptation capability for wildlife in the face of rapidly changing environments [26]. To our knowledge, our study was the first to investigate the bacterial composition of the pronghorn gut microbiome; thus, more data is needed from pronghorn occupying a broader geographic area before conclusions can be made as to whether the core gut microbiome we observed in winter in the Red Desert of south-central Wyoming is similar across the range of pronghorn, or unique to this population. Our study found differing gut microbiome composition in various study areas; if future studies confirm pronghorn gut microbiomes continue to differentiate with increasing habitat differences or geographic distances, pronghorn may be exhibiting locally adapted gut microbiomes. On the other hand, if future studies find microbes in diverse populations are more constrained, it may show there is little plasticity potential in the gut microbiome composition of pronghorn, potentially due to similarity in host diet composition, host genetics, or both. Furthermore, our study shows that movement barriers, such as I-80, may limit a population's ability to exchange microbes with other populations or novel environments, potentially decreasing advantages conferred by microbiome plasticity.

In either scenario, understanding differences or similarities in pronghorn gut microbiomes in distinct habitats may have important implications as wildlife managers increasingly recognize the value of microbial tools for future management. Scientists are discussing the need to consider microbiome composition when reintroducing or translocating animals [19, 23, 27]. The potential value of probiotic treatments for wildlife species has been discussed [19, 28], with some strains of bacteria being evaluated as wildlife gut probiotics [31]. One study identified potential taxa that could serve as bio-indicators for mule deer health [83]. If effective, tools based on locally adapted or core gut microbes in any of these capacities, such as increasing translocation success, serving as a beneficial probiotic to increase herd health, or being used as a biomarker for management, will be valuable not only in pronghorn but also in more vulnerable species. We believe that understanding the gut microbiome composition in pronghorn across their range will prove useful as we attempt to manage populations in a time of growing human disturbance and environmental change and our study provides a baseline understanding of the gut microbiome composition in these iconic herbivores.

## Supporting information

**S1 Table. Pronghorn metadata.**
(CSV)

**S1 Fig. Rarefaction curve.** Dotted line shows the chosen rarefaction point of 4936 reads per sample. Samples are color coded by study area, so we could be assured samples dropped in the rarefaction step were not all from the same location.
(TIF)

**S2 Fig. Pronghorn microbiome composition represented by family relative abundance and grouped by body condition.** Family relative abundance is grouped and visualized by ss-

ligament (measured in inches of depression). Top 15 families depicted make up 95.295% of the assigned amplicon sequence variants (ASVs) present. Larger values for ss-ligament represent leaner animals.
(TIF)

**S3 Fig. Pronghorn microbiome composition represented by family relative abundance and grouped by capture period.** Family relative abundance is grouped and visualized by capture period. Top 15 families depicted make up 95.295% of the assigned amplicon sequence variants (ASVs) present.
(TIF)

**S4 Fig. Pronghorn microbiome composition represented by family relative abundance and grouped by location relative to Interstate-80.** Family relative abundance is grouped and visualized by location relative to Interstate 80. Top 15 families depicted make up 95.295% of the assigned amplicon sequence variants (ASVs) present.
(TIF)

**S5 Fig. Pronghorn microbiome composition for individual samples represented by family relative abundance.** Top 15 families present in the pronghorn microbiome for each individual animal's sample. The top 15 families depicted make up 95.295% of the assigned amplicon sequence variants (ASVs) present (Table 1).
(TIF)

**S1 Appendix. Other tables.** Includes: Table A. Pronghorn captures by time period and study area. Table B. Read count remaining after the various steps in the DADA 2 pipeline in QIIME process. Table C. Pronghorn metrics by study area.
(DOCX)

**S2 Appendix. Additional alpha diversity analyses.** Includes: Table D. Kruskal-Wallis tests of three alpha diversity metrics of pronghorn gut microbiome among discrete pronghorn metrics for animals north and south of I-80. Table E. Values for alpha diversity metrics for sub-groups of pronghorn. Table F. Spearman's rank correlations ($r_s$) between alpha diversity measures and continuous pronghorn life history metrics in animals north and south of I-80. Table G. Models for Observed Richness.
(DOCX)

**S3 Appendix. Single metric PERMANOVAS.** Includes: Table H. Results of single metric PERMANOVA tests. Table I. Results of log transformed single metric PERMANOVA tests.
(DOCX)

**S4 Appendix. PERMANOVAS in single study areas.** Includes: Table J. PERMANOVAS within the Baggs Study Area, n = 36 animals. Table K. PERMANOVAS within the Bitter Creek Study Area, n = 41 animals. Table L. PERMANOVAS within the CDC Study Area, n = 22 animals. Table M. PERMANOVAS within the Red Desert Study Area, n = 35 animals.
(DOCX)

## Acknowledgments

We thank E. Hayden, M. Matocq, R. Liu, and L. Richards for overall mentorship of our project. We thank S. Hudon and the Hayden lab for coordinating sample processing with the Knight lab and the Center for Microbiome Innovation at UC San Diego. The Knight lab at the UC San Diego Center for Microbiome Innovation performed sample extractions and library preparation using protocols and primers published on the Earth Microbiome Project website

(https://earthmicrobiome.org/protocols-and-standards/16s/). This publication includes data generated at the UC San Diego IGM Genomics Center. We thank members of the Boise State University Microbiome Hub for their comradery, advice, and thoughtful discussions during the analysis and writing process. We thank Jonas Frankel-Bricker and Olivia Rodriguez for sharing their codes for analyzing 16-S metabarcoding in R. We thank M. Read from the Rawlins Field Office of the Bureau of Land Management, T. Mong from Wyoming Game and Fish Department, K. Monteith from the University of Wyoming, and Native Range Capture Services for logistical and field support to capture pronghorn to collect biological samples. We thank M. Miller from the Wyoming State Veterinary Laboratory at the University of Wyoming for analyzing blood samples to ascertain disease status.

## Author Contributions

**Conceptualization:** Courtney E. Buchanan, Stephanie J. Galla, Jennifer S. Forbey, Jeffrey L. Beck.

**Data curation:** Courtney E. Buchanan, Adele K. Reinking.

**Formal analysis:** Courtney E. Buchanan, Stephanie J. Galla, Mario E. Muscarella.

**Funding acquisition:** Jennifer S. Forbey, Adele K. Reinking, Jeffrey L. Beck.

**Investigation:** Adele K. Reinking.

**Methodology:** Jeffrey L. Beck.

**Project administration:** Jennifer S. Forbey, Jeffrey L. Beck.

**Resources:** Jennifer S. Forbey, Jeffrey L. Beck.

**Supervision:** Stephanie J. Galla, Mario E. Muscarella, Jennifer S. Forbey, Jeffrey L. Beck.

**Visualization:** Courtney E. Buchanan, Adele K. Reinking.

**Writing – original draft:** Courtney E. Buchanan, Stephanie J. Galla, Jeffrey L. Beck.

**Writing – review & editing:** Courtney E. Buchanan, Stephanie J. Galla, Mario E. Muscarella, Jennifer S. Forbey, Adele K. Reinking, Jeffrey L. Beck.

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
