## [Decision Letter · Decision Letter 0]

11 Dec 2023

PONE-D-23-37173Relating gut microbiome composition and life history metrics for pronghorn (*Antilocapra americana *) in the Red Desert, WyomingPLOS ONE

Dear Dr. Buchanan,

Thank you for submitting your manuscript to PLOS ONE. After careful consideration, we feel that it has merit but does not fully meet PLOS ONE’s publication criteria as it currently stands. Therefore, we invite you to submit a revised version of the manuscript that addresses the points raised during the review process.

ACADEMIC EDITOR:Your manuscript has been reviewed. I want you to respond to the reviewer's comments and revise the manuscript.Please submit your revised manuscript by Jan 25 2024 11:59PM. If you will need more time than this to complete your revisions, please reply to this message or contact the journal office at plosone@plos.org. Please include the following items when submitting your revised manuscript:A rebuttal letter that responds to each point raised by the academic editor and reviewer(s). You should upload this letter as a separate file labeled 'Response to Reviewers'.A marked-up copy of your manuscript that highlights changes made to the original version. You should upload this as a separate file labeled 'Revised Manuscript with Track Changes'.An unmarked version of your revised paper without tracked changes. You should upload this as a separate file labeled 'Manuscript'.

We look forward to receiving your revised manuscript.

Kind regards,

Faham Khamesipour, Ph.D.

Academic Editor

PLOS ONE

Journal Requirements:

2. Thank you for stating the following financial disclosure: "The sequencing data collection and analysis and preparation of the manuscript was supported by the U.S. National Science Foundation (https://www.nsf.gov/): Grants OIA-1826801 to J. Forbey and OIA-1738865 to E. Hayden. 

Funding for fecal sample and pronghorn field data collection was funded with a combination of grants and financial support from agency, university, and industry partners listed below. Funds were awarded to J. Beck between 2009 and 2016 for previous pronghorn research. Our research used legacy samples that were collected during this previous research project. 

Funders did not play a role in study design, data collection and analysis, decision to publish, or preparation of the manuscript with the exception of logistical and field support provided by BLM and WGFD staff mentioned in acknowledgements during the sample collection phase of the project. 

Anadarko Petroleum Corporation (acquired by Occidental Petroleum in 2019 https://www.oxy.com/). Institution award number: 1002418

Black Diamond Minerals LLC (Merged into MRD Operating, LLC in 2014). Institution award number: 1001671

British Petroleum North America (https://www.bp.com/). No institution award number or grant information available. 

Bureau of Land Management-Rawlins Field Office (https://www.blm.gov/office/rawlins-field-office). Sponsor ID: L16AC00156 and L09AC15996

Devon Energy (https://www.devonenergy.com/). Institution award number: 1002087

Linn Energy (https://linnenergy.com/). Institution award number: 1002724

Memorial Resource Development (acquired by Range Resources Corporation in 2016 https://www.rangeresources.com/). Institution award number: 1002818

Samson Resources (https://www.samsonco.com/index.aspx). Institution award number: 1001972

Warren Resources, Incorporated (https://warrenresources.com). No institution award number or grant information available.

Wyoming Game and Fish Department (https://wgfd.wyo.gov/). No institution award number or grant information available.

Wyoming Governor’s Big Game License Coalition (https://wgfd.wyo.gov/Apply-or-Buy/Commissioner-and-Governor-Licenses/Governors-Big-Game-License). No institution award number or grant information available.

University of Wyoming: Department of Ecosystem Science and Management(https://www.uwyo.edu/esm/index.html), Office of Academic Affairs (https://www.uwyo.edu/acadaffairs/index.html), and Wyoming Reclamation and Restoration Center (https://www.uwyo.edu/wrrc/index.html),. No institution award number or grant information available."

Reviewers' comments:

Reviewer's Responses to Questions

**Comments to the Author**

1. Is the manuscript technically sound, and do the data support the conclusions?

Reviewer #1: Yes

Reviewer #2: Partly

Reviewer #3: Yes

2. Has the statistical analysis been performed appropriately and rigorously? 

Reviewer #1: Yes

Reviewer #2: No

Reviewer #3: Yes

3. Have the authors made all data underlying the findings in their manuscript fully available?

Reviewer #1: Yes

Reviewer #2: No

Reviewer #3: Yes

4. Is the manuscript presented in an intelligible fashion and written in standard English?

Reviewer #1: Yes

Reviewer #2: Yes

Reviewer #3: Yes

5. Review Comments to the Author

Reviewer #1: Buchanan et al provide the first characterization of pronghorn microbiome using 16S microbiome metabarcoding of fecal samples. They also found differences in diversity and small differences in composition related with sampling are time. I think this study is an important resource for future research on the impacts of gut microbiome in this species fitness and conservation. The manuscript is generally well written, and structure and the methods are appropriate to address the questions set by the authors, and authors conclusions seem adequate given the results obtained. Nevertheless, the authors should consider some minor revisions that I outline below.

Line 31: winters of

Line 42: myriad of

Line 44: Citations missing. These could be just some of the most emblematic examples mentioned by the authors.

Line68: This sentence seems like a sum of the ideas that have been developed in this paragraph but at the same time gives the sensation that needs to be developed even further. I think it should be placed in the beginning of the paragraph.

Line 76: All shrubs or just Artemisia spp.?

Line 77: Reference missing

Line 79: In the abstract the authors mention that the pronghorn is a facultative Artemisia specialist. I think dietary preferences should be mentioned in the introduction as well because they provide an idea of the function that microbiome must play in sagebrush fitness and the dependency that pronghorn might have in the integrity of sagebrush steppe biome.

Line 113: This is the first time that the authors mention serum disease. They should introduce the impact that it has on pronghorn populations.

Line 128: For congruence the authors should add the scientific names for the greater sage-grouse and mule deer.

Line 138: Why females only?

Line 140: Why 159? What happened to the remaining eight?

Line 242: In the material and methods these two age measures. How did the authors correct the age estimates?

Line 359: The best explanatory variable?

General results: Since significant differences between sides of I-80 and several life history measures showed interaction with study are, these might indicate that these have opposite effects on different sides if I-80. To verify these, I would test if there were differences in diversity and correlation with life-history variables within populations of each side of I80.

Figures: The authors need to provide higher definition pictures. For figure 7B, please increase the point size. It is hard to distinguish between circles and triangles.

Line 449: What did the authors mean by not being able to include the capture period? DO they mean that they were not able to include all seasons?

Line 468: I do not see any relationship since p<0.1 for before and after interactions. The chosen alpha is already larger than the usual 0.05, so I think saying that p values close to 0.2 indicate a subtle relation without any other indication is not correct. For me the interesting part is the fact that you find a significant interaction between study-area and Ss-ligament. This indicates that the latter might have different effects in different sample localities, which should be explored.

Reviewer #2: This is largely a descriptive study of the fecal microbiome of the pronghorn (Antilocapra americana). Generally the article is well written, however the statistical analyses carried out by the authors are weak and flawed. The authors could also do much more with the data they have.

Main concerns

The authors mention that they included blanks as negative controls and they have used the “decontam” R package to control for potential contaminants in their positive samples. This is great, but they do not mention how many blanks were included for sequencing, nor whether these negative controls were field blanks, extraction blanks or PCR blanks (or a combination of these). This needs to be clearly indicated.

For alpha diversity analyses, the authors have only made pairwise Wilcoxon tests. Yet they have several variables that may reasonably impact diversity. Testing for them independently does not account for the effect of all variables when testing for the effect of a single variable. This can be done easily using a multiple regression method that accounts for the non-normality of their data such as general linear models with the appropriate error distributions.

For beta diversity analyses the authors mention that the non-rarefied data was transformed to relative abundance. However they do not explain how this was done and a range of methods are available (see for example, but there are many others, the following studies: https://www.frontiersin.org/articles/10.3389/fmicb.2017.02224/full, https://www.frontiersin.org/articles/10.3389/fmicb.2021.727398/full, https://bmcbioinformatics.biomedcentral.com/articles/10.1186/s12859-023-05205-3,https://cdnsciencepub.com/doi/10.1139/cjm-2015-0821). This needs to be clarified. It is also now recommended to rarefy the dataset even when using recommended compositional methods (see https://www.biorxiv.org/content/10.1101/2023.06.23.546312v1.abstract, https://www.biorxiv.org/content/10.1101/2023.06.23.546313v1.abstract).

The significance of ss-ligament is briefly mentioned in the introduction but the authors fail to describe how it is estimated in the methods. This needs to be clarified.

The authors write in lines 223-225: “ In the second phase, for each metric of interest, we chose one of the related factors for input into a model, avoiding inputting multiple factors that represented the same metadata metric.” To me it is not clear at all what the authors mean by this nor is it an understandable statistical approach.

Lines 254-256: What do the authors mean by “dispersion”. Do they in fact mean composition?

The authors frequently report p-values above the traditional threshold of 0.05 as being significant (lines 370 and 386). What is their threshold of significance?I would recommend revising these claims of statistical support and strongly suggest avoiding reporting a non-significant p-value as significant.

I understand that this is a descriptive study, however testing interaction terms with little biological premise makes little sense and is not recommended (see for instance https://elifesciences.org/articles/48175). I suggest to only test and report interactions if there is a reasonable biological reason to test this interaction.

Line 387: 8.8.% is actually quite a strong effect, especially in microbiome research where many variables are likely to affect community structure. It is not clear what the authors mean by “adjusted”, i.e. how is 8.8% assessed and how does that contrast with their estimate of 3.1%?

Overall, the analytical approach is not only flawed but it gives the impression that the authors have done very little effort to do a well thought out study design and statistical approach, even when considering the descriptive objectives of the study.

The authors do not discuss the lack of impact of disease status (Epizootic Hemorrhagic Disease and Bluetongue Virus) despite testing for them in their analyses. Whilst I understand that ultimately within the strong limitations of their study, they found no effect, it is surprising that these results were not put in the broader context of the strong literature to date on the link between disease and gut microbiomes.

Reviewer #3: Overall well executed study and well written manuscript, I only have minor comments.

The distinction between host microbiome and specifically the gut microbiome is somewhat inconsistent/not well distinguished. I think using "gut microbiome" when speaking to the gut microbiome specifically will provide more clarity to the reader. Perhaps introduce/define the gut microbiome in the introduction as well.

Line 76/77 does the gut microbiome contribute to herbivore physiological adaptations to chemically defended shrubs? Providing a more direct link between the landscape and the study species through the gut microbiome would strengthen the argument for why this study is important.

Are any of the individuals assessed for body condition considered "unhealthy"?

Reporting similar bacterial communities across grouped samples at the phyla, class, order, and family level is not as compelling as perhaps looking at genera. Further, it may undermine the importance of lower abundance bacteria that play a significant role in host health. Did the authors consider using linear discrimination analysis effect size (LEfSe) to identify any ASVs that were significantly enriched between grouped samples?

The chosen significant p value (p<0.1) is a bit weaker than significant p values used in similar studies (p<0.05).

Figure 2. Having the outline the same color (black) as "additional Phyla" is confusing, use a different color or remove outlines. Is the relative abundance cutoff for "additional phyla" given in the manuscript? If we can't see majority of phyla described in this figure, then why are they separate from "additional phyla"? Would you be able to see them without the black outlines? Same comment for figures 3-5.

6. PLOS authors have the option to publish the peer review history of their article (what does this mean?). If published, this will include your full peer review and any attached files.

Reviewer #1: No

Reviewer #2: No

Reviewer #3: No

---

## [Author Response · Author response to Decision Letter 0]

2 Apr 2024

To the Editors at PLoS,

We are submitting a revision for our manuscript titled “Relating gut microbiome composition and life history metrics for pronghorn (Antilocapra americana) in the Red Desert, Wyoming”. We appreciate the comments of all three reviewers and have done our best to address their concerns with our manuscript. We feel the comments have helped us to improve the manuscript. As requested, we have uploaded a track-change version showing the adjustments we have made as well as a clean version of the revised manuscript. Included below are the reviewer comments in followed by the authors’ responses to each comment. In addition we have addressed the additional requirements that were listed in our decision email and responded in a similar manner following the reviewer comments. We re-uploaded new versions of figures and Supporting Information Files as there were changes in these files. 

Thank you for considering our revised manuscript and we look forward to your response. 

Sincerely,

Courtney E. Buchanan, PhD Candidate

Department of Ecosystem Science and Management

University of Wyoming

Laramie, Wyoming 82071

Reviewer #1: Buchanan et al provide the first characterization of pronghorn microbiome using 16S microbiome metabarcoding of fecal samples. They also found differences in diversity and small differences in composition related with sampling are time. I think this study is an important resource for future research on the impacts of gut microbiome in this species fitness and conservation. The manuscript is generally well written, and structure and the methods are appropriate to address the questions set by the authors, and authors conclusions seem adequate given the results obtained. Nevertheless, the authors should consider some minor revisions that I outline below.

 Line 31: winters of

Author Response: This was changed (line 30).

 Line 42: myriad of

Author Response: This was changed (line 41).

Line 44: Citations missing. These could be just some of the most emblematic examples mentioned by the authors.

Author Response: Citation 1 was also meant to be the citation for this sentence, so we added a citation for this sentence as well to make it more clear (line 44). This paper we cite, in addition to discussing the theory of holobionts, gives numerous examples of studies that use this principle.

Line68: This sentence seems like a sum of the ideas that have been developed in this paragraph but at the same time gives the sensation that needs to be developed even further. I think it should be placed in the beginning of the paragraph.

Author Response: We moved this into the first sentence as suggested, and incorporated this idea into the opening sentence (line 59-62). We also altered the final sentence of this paragraph to maintain flow to the topic of the next paragraph.

Line 76: All shrubs or just Artemisia spp.?

Author Response: Both. This is now clarified in lines 78-81. However, we focus much of the continuing discussion of plant toxins and the capability of animals to consume these plants specifically on sagebrush (lines 81-86).

Line 77: Reference missing

Author Response: We added references exhibiting that few animals that are able to consume large amounts of sagebrush as well as documenting the abilities of these other sagebrush specialists to consume sagebrush plants. See lines 81-86.

Line 79: In the abstract the authors mention that the pronghorn is a facultative Artemisia specialist. I think dietary preferences should be mentioned in the introduction as well because they provide an idea of the function that microbiome must play in sagebrush fitness and the dependency that pronghorn might have in the integrity of sagebrush steppe biome.

Author Response: We added this information into the introduction along with the topics discussing other sagebrush specialists. See lines 86-89.

Line 113: This is the first time that the authors mention serum disease. They should introduce the impact that it has on pronghorn populations.

Author Response: This was initially cited within the sentence detailing environmental factors affecting pronghorn populations (see citation #77), however we expanded upon this in lines 103- 106.

Line 128: For congruence the authors should add the scientific names for the greater sage-grouse and mule deer.

Author Response: We added the scientific names for these species to this sentence (line 145). Because these species were already mentioned in the above text we followed PLOS ONE guidelines and included the first letter of the genus name and the full species name.

Line 138: Why females only?

Author Response: We had no specific objective in only sampling females. The samples for our study were legacy samples gathered in 2013 and 2014 during pronghorn captures for a previous study. We added lines 137-138 to the methods to make this clearer. For that study only female animals were captured, so we only had access to legacy female samples for our study. In these lines the reader is directed to published literature that details this previous study. We have also included these references below: 

• Reinking AK, Smith KT, Mong TW, Read MJ, Beck JL. Across scales, pronghorn select sagebrush, avoid fences, and show negative responses to anthropogenic features in winter. Ecosphere. 2019;10(5). doi: 10.1002/ecs2.2722

• Reinking AK, Smith KT, Monteith KL, Mong TW, Read MJ, Beck JL. Intrinsic, environmental, and anthropogenic factors related to pronghorn summer mortality. J Wildl Manage. 2018;82(3):608–17. doi: 10.1002/jwmg.21414

Line 140: Why 159? What happened to the remaining eight?

Author Response: Some samples were removed from the analysis due to ambiguous metadata (lines 219-220).

Line 242: In the material and methods these two age measures. How did the authors correct the age estimates?

Author Response: This is detailed in the supplemental info, but we added it to the methods in the main manuscript as well. See lines 164-167.

 Line 359: The best explanatory variable?

Author Response: Yes. This has been changed to ‘best’ (line 405).

General results: Since significant differences between sides of I-80 and several life history measures showed interaction with study are, these might indicate that these have opposite effects on different sides if I-80. To verify these, I would test if there were differences in diversity and correlation with life-history variables within populations of each side of I80.

Author Response: In regards to diversity, we did find that there was a significant difference (at the alpha 0.10 level) in alpha diversity between the two sides of I-80 for both Shannon Diversity Index and Observed Richness (See Table 2). 

As suggested, we ran additional correlations and Kruskal-Wallis tests between pronghorn life history variables and alpha diversity indices on data subsets of pronghorn north and south of I-80. We found some subtle differences in these relationships between the two populations, however we felt that these findings do not change the overall story we see in our data. We detailed the results in the supporting information (see Appendix 2), as described in lines 238-241 and 347-349.

Figures: The authors need to provide higher definition pictures. For figure 7B, please increase the point size. It is hard to distinguish between circles and triangles.

Author Response: We note that in the pdf download created by the PLOS submission portal the figures appear grainy. We assume this to be a relic of the website that creates the viewable pdf. The individual figure files have been created according to the PLOS standard for DPI and were uploaded through PACE to ensure the standards were met. We increased the point size in Figure 7B, as well as we changed the circles and triangles in 7A and 7B to circles and plus signs to make the shapes more distinguishable.

Line 449: What did the authors mean by not being able to include the capture period? DO they mean that they were not able to include all seasons?

Author Response: The fecal samples that were collected in February were missing some pronghorn metrics that were collected during other capture periods. For our analysis with multiple variables, we needed to remove all samples with missing data in any of our metrics of interest, thus all February samples were removed. In addition, fecal samples from one of our study areas were only collected during the third capture period, meaning that study area was confounded with capture period. Due to these two issues, we did not feel justified to include a capture period metric in multivariate models, though our single metric models indicated it was potentially explanatory, which is also supported by many of our alpha diversity analyses. This is explained in Lines 28-34 in S3 Appendix. Readers are directed to this appendix in lines 270-271. 

Line 468: I do not see any relationship since p<0.1 for before and after interactions. The chosen alpha is already larger than the usual 0.05, so I think saying that p values close to 0.2 indicate a subtle relation without any other indication is not correct. For me the interesting part is the fact that you find a significant interaction between study-area and Ss-ligament. This indicates that the latter might have different effects in different sample localities, which should be explored.

Author Response: It is the understanding of the authors that once an interaction is performed, the p-value of interest to be interpreted is no longer the one of the initial variable (in this case the 0.191) but rather the one represented by the interaction (in this case the 0.035). The subtle effect mentioned is due to the fact that ss-ligament has an interaction with the study area metric, but was not significant on its own in the model with multiple variables. We also provided more clarification to this point in the Discussion (lines 519 – 522). 

We give the caveat in lines 524-527 that “while some of our analyses provide support for subtle gut microbiome differences in animals of varying body condition, our results do not conclusively show a clear link between gut microbiome composition and measures for body condition.” We did explore the effect of ss-ligament in different study areas, by sub-setting our data by study area and running a PERMANOVA with the same metrics on each subset. We did not see any significant results for ss-ligament when we did this, potentially because our sample size was now smaller for individual study areas than our grouped data. Those results are detailed in lines (423 - 424) and we added the outputs as well to Appendix 4.

Reviewer #2: This is largely a descriptive study of the fecal microbiome of the pronghorn (Antilocapra americana). Generally the article is well written, however the statistical analyses carried out by the authors are weak and flawed. The authors could also do much more with the data they have.

 Main concerns

The authors mention that they included blanks as negative controls and they have used the “decontam” R package to control for potential contaminants in their positive samples. This is great, but they do not mention how many blanks were included for sequencing, nor whether these negative controls were field blanks, extraction blanks or PCR blanks (or a combination of these). This needs to be clearly indicated.

Author Response: We contacted the lab that ran the samples and they confirmed these were extraction blanks. We added information about the blanks in lines 196-198.

For alpha diversity analyses, the authors have only made pairwise Wilcoxon tests. Yet they have several variables that may reasonably impact diversity. Testing for them independently does not account for the effect of all variables when testing for the effect of a single variable. This can be done easily using a multiple regression method that accounts for the non-normality of their data such as general linear models with the appropriate error distributions.

Author Response: The objectives of our alpha diversity analyses were to explore whether different groups of animals may have differing means for alpha diversity values, rather than the approach of regression which would be to look at the specific effect of each predictor on alpha diversity. We chose to look at these variables independently as many of the metrics we wanted to look at were confounded with one another. This was either due to when and where animals were captured, or due to metrics that were correlated with one another (e.g., our two body fat measures). Investigating these factors individually allowed us to better identify which groups of animals may have differing alpha diversity in their gut microbiomes.

Nevertheless, to investigate this further we built a series of models to evaluate the effects of the predictor variables together on alpha diversity as measured by observed richness. Overall, the results were similar to what we saw with our correlations and pairwise tests. We saw an effect of capture period in these models, but did not measure a significant effect for other pronghorn metrics on alpha diversity. As this supported what we found in our other analyses, we did not investigate this avenue further. See lines (241 – 243) and Appendix 2 for more detail and results. 

For beta diversity analyses the authors mention that the non-rarefied data was transformed to relative abundance. However they do not explain how this was done and a range of methods are available (see for example, but there are many others, the following studies: https://www.frontiersin.org/articles/10.3389/fmicb.2017.02224/full, https://www.frontiersin.org/articles/10.3389/fmicb.2021.727398/full, https://bmcbioinformatics.biomedcentral.com/articles/10.1186/s12859-023-05205-3,https://cdnsciencepub.com/doi/10.1139/cjm-2015-0821). This needs to be clarified.

Author Response: We clarified this in lines 245-247. In short, we did a simple transformation to relative abundance by taking the number of read counts for a particular taxa and dividing by the total number of reads for each sample. We also did a log transformation of our abundance data to investigate whether results would be different when represented on the logarithmic scale. These analyses are detailed in the Appendix 3 and we direct the reader to the appendix for more information (see lines 247-251).

 It is also now recommended to rarefy the dataset even when using recommended compositional methods (see https://www.biorxiv.org/content/10.1101/2023.06.23.546312v1.abstract, https://www.biorxiv.org/content/10.1101/2023.06.23.546313v1.abstract).

We would like to thank the reviewer for their comments and the informative references that were provided. However, the methods we have employed are supported by many examples in the literature. While some authors have recommended rarefaction for both alpha and beta diversity, others have statistically shown that not rarefying relative abundance data is a more robust approach: See https://www.frontiersin.org/journals/microbiology/articles/10.3389/fmicb.2019.02407/full. As such, we chose to use relative abundance data without rarefaction for our beta diversity analysis.

Much of the literature you have shared also suggests a log-transformation of the relative abundance. We applied log transformation to account for the high dominance of a few ASVs, however we found the results to be very similar to those produced with the non-log-transformed data (see Appendix 3). Therefore, we chose to use the non-log transformed data for simplicity of interpretation of the data.

The significance of ss-ligament is briefly mentioned in the introduction but the authors fail to describe how it is estimated in the methods. This needs to be clarified.

Author Response: We explain how the ss-ligament is determined during animal capture in lines 168-170. This is also further explained in lines 289-291 when discussing the range of values that were recorded for this measurement.

The authors write in lines 223-225: “ In the second phase, for each metric of interest, we chose one of the related factors for input into a model, avoiding inputting multiple factors that represented the same metadata metric.” To me it is not clear at all what the authors mean by this

---

## [Decision Letter · Decision Letter 1]

28 May 2024

PONE-D-23-37173R1Relating gut microbiome composition and life history metrics for pronghorn (*Antilocapra americana *) in the Red Desert, WyomingPLOS ONE

Dear Dr. Buchanan,

Thank you for submitting your manuscript to PLOS ONE. After careful consideration, we feel that it has merit but does not fully meet PLOS ONE’s publication criteria as it currently stands. Therefore, we invite you to submit a revised version of the manuscript that addresses the points raised during the review process.

**ACADEMIC EDITOR:**I have suggested some minor revisions to enhance the clarity and impact of your manuscript. Please find the reviewer comment. I kindly request that you address these minor revisions and submit the revised manuscript. ==============================Please submit your revised manuscript by Jul 12 2024 11:59PM. If you will need more time than this to complete your revisions, please reply to this message or contact the journal office at plosone@plos.org. Please include the following items when submitting your revised manuscript:A rebuttal letter that responds to each point raised by the academic editor and reviewer(s). You should upload this letter as a separate file labeled 'Response to Reviewers'.A marked-up copy of your manuscript that highlights changes made to the original version. You should upload this as a separate file labeled 'Revised Manuscript with Track Changes'.An unmarked version of your revised paper without tracked changes. You should upload this as a separate file labeled 'Manuscript'.If applicable, we recommend that you deposit your laboratory protocols in protocols.io to enhance the reproducibility of your results. Protocols.io assigns your protocol its own identifier (DOI) so that it can be cited independently in the future. For instructions see: https://journals.plos.org/plosone/s/submission-guidelines#loc-laboratory-protocols. Additionally, PLOS ONE offers an option for publishing peer-reviewed Lab Protocol articles, which describe protocols hosted on protocols.io. Read more information on sharing protocols at https://plos.org/protocols?utm_medium=editorial-email&utm_source=authorletters&utm_campaign=protocols.

We look forward to receiving your revised manuscript.

Kind regards,

Faham Khamesipour, Ph.D.

Academic Editor

PLOS ONE

Journal Requirements:

Additional Editor Comments:

I have suggested some minor revisions to enhance the clarity and impact of your manuscript. Please find the reviewer comment. I kindly request that you address these minor revisions and submit the revised manuscript.

Reviewers' comments:

Reviewer's Responses to Questions

**Comments to the Author**

1. If the authors have adequately addressed your comments raised in a previous round of review and you feel that this manuscript is now acceptable for publication, you may indicate that here to bypass the “Comments to the Author” section, enter your conflict of interest statement in the “Confidential to Editor” section, and submit your "Accept" recommendation.

Reviewer #1: All comments have been addressed

2. Is the manuscript technically sound, and do the data support the conclusions?

Reviewer #1: Yes

3. Has the statistical analysis been performed appropriately and rigorously? 

Reviewer #1: Yes

4. Have the authors made all data underlying the findings in their manuscript fully available?

Reviewer #1: Yes

5. Is the manuscript presented in an intelligible fashion and written in standard English?

Reviewer #1: Yes

6. Review Comments to the Author

Reviewer #1: In the present version of the manuscript the authors have addressed all my previous suggestions. In the moment I only have a few minor comments:

Line 137: Why were these study areas chosen? Do they correspond to areas with higher known pronghorn densities? Were they chosen to test the impact of I80? Are they part of a program? I know it is not crucial information, but it may help the reader to better understand the framework in which the study was developed.

Line 484: I think the authors should be careful when mentioning season effect since the sampling is highly biased. For example, most samples come from November 2013 and for November 2014, most of the individuals the most represented locality was not included in the other periods. Thus, this pattern can be an artifact which should be mentioned.

Line 520: How did the authors observe that there was a subtle relationship, even though it was not significant?

7. PLOS authors have the option to publish the peer review history of their article (what does this mean?). If published, this will include your full peer review and any attached files.

Reviewer #1: No

---

## [Author Response · Author response to Decision Letter 1]

7 Jun 2024

Reviewer #1: In the present version of the manuscript the authors have addressed all my previous suggestions. In the moment I only have a few minor comments:

Line 137: Why were these study areas chosen? Do they correspond to areas with higher known pronghorn densities? Were they chosen to test the impact of I80? Are they part of a program? I know it is not crucial information, but it may help the reader to better understand the framework in which the study was developed.

Author Response: The samples for our study were legacy samples gathered in 2013 and 2014 during pronghorn captures for a previous study, and thus the study areas reflected the objectives of this earlier study. This initial study was conducted to better understand how environmental and intrinsic factors and anthropogenic disturbances affected pronghorn survival and seasonal habitat selection by pronghorn populations in the Red Desert of south-central Wyoming. The study locations were chosen as known areas of pronghorn habitation corresponding to 5 Wyoming Game and Fish Department hunt areas which also had the requisite oil and gas infrastructure for that study. These areas were not specifically chosen to test impacts of I-80 but were chosen to look at energy development infrastructure as well as a range of environmental and anthropogenic features. We added lines 137-144 to the description of study areas to make this clearer to the reader. In this section the reader is also directed to published literature that details this previous study for more information. We have also included these references below: 

• Reinking AK, Smith KT, Mong TW, Read MJ, Beck JL. Across scales, pronghorn select sagebrush, avoid fences, and show negative responses to anthropogenic features in winter. Ecosphere. 2019;10(5). doi: 10.1002/ecs2.2722

• Reinking AK, Smith KT, Monteith KL, Mong TW, Read MJ, Beck JL. Intrinsic, environmental, and anthropogenic factors related to pronghorn summer mortality. J Wildl Manage. 2018;82(3):608–17. doi: 10.1002/jwmg.21414

Line 484: I think the authors should be careful when mentioning season effect since the sampling is highly biased. For example, most samples come from November 2013 and for November 2014, most of the individuals the most represented locality was not included in the other periods. Thus, this pattern can be an artifact which should be mentioned.

We understand that the sampling in our study was not perfect due to the legacy sample nature of the study and thus there is bias in the number of samples from each season, as well as when samples were collected from certain areas. We did address these issues briefly in the Discussion in lines 498-499 and 509-511 and added further explanations to address this in our Discussion (lines 487-491). While the imperfect nature of the legacy study design does not allow us to make concrete claims about the season of capture influencing the microbiome, we feel that its continued appearance in our analyses warrants mentioning its potential influence and the need to investigate this effect further in studies more carefully designed for that purpose. 

Line 520: How did the authors observe that there was a subtle relationship, even though it was not significant?

We adjusted the wording of this section (lines 529-538) to make our point clearer. The point we are hoping to get across is that we had some mixed results leading to us not having a clear idea of the relationship between ss-ligament and microbiome. Ss-ligament was significant when tested in our initial individual PERMANOVA tests without other predictors but was not significant when tested in the combined model with other predictors; however, it did have a significant interaction with study area in the combined model. Therefore, while there was not a clear relationship with ss-ligament and the gut microbiome, there is a potential context-dependent relationship. Thus, while we were not able to show a clear relationship in our study, there is good reason for it to be investigated further in future studies. We hope this adjustment makes this point clearer.

---

## [Decision Letter · Decision Letter 2]

24 Jun 2024

Relating gut microbiome composition and life history metrics for pronghorn (*Antilocapra americana *) in the Red Desert, Wyoming

PONE-D-23-37173R2

Dear Dr. Buchanan,

We’re pleased to inform you that your manuscript has been judged scientifically suitable for publication and will be formally accepted for publication once it meets all outstanding technical requirements.

Kind regards,

Faham Khamesipour, Ph.D.

Academic Editor

PLOS ONE

Additional Editor Comments (optional):

ready for publication

Reviewers' comments:

Reviewer's Responses to Questions

**Comments to the Author**

1. If the authors have adequately addressed your comments raised in a previous round of review and you feel that this manuscript is now acceptable for publication, you may indicate that here to bypass the “Comments to the Author” section, enter your conflict of interest statement in the “Confidential to Editor” section, and submit your "Accept" recommendation.

Reviewer #1: All comments have been addressed

Reviewer #3: All comments have been addressed

2. Is the manuscript technically sound, and do the data support the conclusions?

Reviewer #1: Yes

Reviewer #3: Yes

3. Has the statistical analysis been performed appropriately and rigorously? 

Reviewer #1: Yes

Reviewer #3: Yes

4. Have the authors made all data underlying the findings in their manuscript fully available?

Reviewer #1: Yes

Reviewer #3: Yes

5. Is the manuscript presented in an intelligible fashion and written in standard English?

Reviewer #1: Yes

Reviewer #3: Yes

6. Review Comments to the Author

Reviewer #1: All my previous comments have been addressed. Just a few final points that were not clear to me:

Line 264: This sentence is not clear to me. I think the term “metadata metrics” is creating some confusion. These are the metrics about the animal condition and age, right? If so, just specify that. I also think it is necessary to specify a criterion used for binning the continuous variables.

Line 267: Each metric corresponds to one variable, right? I don’t understand what is there to choose.

Line 290: Were there animals positive for both?

Figures 2 to 5: Probably can be merged into a single compound figure.

Reviewer #3: (No Response)

7. PLOS authors have the option to publish the peer review history of their article (what does this mean?). If published, this will include your full peer review and any attached files.

Reviewer #1: No

Reviewer #3: No

---

## [Editor Report · Acceptance letter]

1 Jul 2024

PONE-D-23-37173R2 

PLOS ONE

Dear Dr. Buchanan, 

I'm pleased to inform you that your manuscript has been deemed suitable for publication in PLOS ONE. Congratulations! Your manuscript is now being handed over to our production team.

Kind regards, 

on behalf of

Dr. Faham Khamesipour 

Academic Editor

PLOS ONE